**Characteristics of total gaseous mercury (TGM) concentrations in an**
**industrial complex in southern Korea: Impacts from local sources**
Yong-Seok Seo[1, 2], Seung-Pyo Jeong[1], Thomas M. Holsen[3], Young-Ji Han[4], Eunhwa Choi[5], Eun
Ha Park[1], Tae Young Kim[1], Hee-Sang Eum[1], Dae Gun Park[1], Eunhye Kim[6], Soontae Kim[6],
Jeong-Hun Kim[7], Jaewon Choi[8], Seung-Muk Yi[1, 2, *]
[1]Department of Environmental Health, Graduate School of Public Health, Seoul National
University, 1 Gwanak, Gwanak-ro, Gwanak-gu, Seoul 151-742, South Korea
[2]Institute of Health and Environment, Seoul National University, 1 Gwanak, Gwanak-ro,
Gwanak-gu, Seoul 151-742, South Korea
[3]Department of Civil and Environmental Engineering, Clarkson University, Potsdam,
NY13699, USA
[4]Department of Environmental Science, Kangwon National University, 192-1, Hyoja-2-dong,
Chuncheon, Kangwondo, 200-701, South Korea
[5]Asian Institute for Energy, Environment & Sustainability, Seoul National University, 1
Gwanak-ro, Gwanak-gu, Seoul 151-742, South Korea
[6]Department of Environmental, Civil and Transportation Engineering, Ajou University,
Woncheon-dong, Yeongtong-gu, Suwon, 443-749, South Korea
[7]Division of Air Pollution Engineering, Department of Climate and Air Quality Research,
National Institute of Environmental Research, Hwangyong-ro 42, Seogu, Incheon, 404-708,
South Korea
[8]University of Pennsylvania, Philadelphia, PA19104, USA
[*]Address correspondence to Dr. Seung-Muk Yi, Graduate School of Public Health, Seoul
National University, 1 Gwanak, Gwanak-ro, Gwanak-gu, Seoul 151-742, South Korea
E-mail) yiseung@snu.ac.kr
Telephone) 82-2-880-2736
Fax) 82-2-745-9104

**Abstract**


Total gaseous mercury (TGM) concentrations were measured every 5 min in Pohang,
Gyeongsangbuk-do, Korea during summer (17 August~23 August 2012), fall (9 October~17
October 2012), winter (22 January ~29 January 2013), and spring (26 March~3 April 2013)
to: 1) characterize the hourly and seasonal variations of atmospheric TGM concentrations, 2)
identify the relationships between TGM and co-pollutants, and 3) identify likely source
directions and locations of TGM using conditional probability function (CPF), conditional
bivariate probability function (CBPF) and total potential source contribution function
(TPSCF).
The TGM concentration was statistically significantly highest in fall ($6.7 \pm 6.4$ ng m$^{-3}$),
followed by spring ($4.8 \pm 4.0$ ng m$^{-3}$), winter ($4.5 \pm 3.2$ ng m$^{-3}$) and summer ($3.8 \pm 3.9$ ng m$^{-3}$
). There was a weak but statistically significant negative correlation between the TGM
concentration and ambient air temperature *(r = -0.08) (p < 0.05).* Although the daytime
temperature ($14.7 \pm 10.0$ ºC) was statistically significantly higher than that in the nighttime
($13.0 \pm 9.8$ ºC) ($p < 0.05$), the daytime TGM concentration ($5.3 \pm 4.7$ ng m$^{-3}$) was statistically
significantly higher than those in the nighttime ($4.7 \pm 4.7$ ng m$^{-3}$) ($p < 0.01$), possibly due to
local emissions related to industrial activities and activation of local surface emission
sources. The observed  $\Delta$TGM/$\Delta$CO was significantly lower than that of Asian long-range
transport, but similar to that of local sources in Korea and in US industrial events suggesting
that local sources are more important than that of long-range transport. CPF, CBPF and
TPSCF indicated that the main sources of TGM were iron and manufacturing facilities, the
hazardous waste incinerators and the coastal areas.
**Keywords**: Total gaseous mercury (TGM); co-pollutant; conditional probability function
(CPF); conditional bivariate probability function (CBPF); total potential source contribution
function (TPSCF)
**1.  Introduction**
Mercury (Hg) is an environmental toxic and bioaccumulative trace metal whose emissions
to the environment have considerably increased due to anthropogenic activities such as
mining and combustion processes (Pirrone et al., 2013; Streets et al., 2011). Hg can be
globally distributed from the sources through atmospheric transport as gaseous elemental
form (Bullock et al., 1998; Mason and Sheu, 2002). However, the origins of atmospheric
mercury are local and regional (Choi et al., 2009) as well as hemispherical and global
(Durnford et al., 2010). In addition to the general background concentration of Hg in the
global atmosphere, local Hg emissions contribute to the Hg burden and it contribute to the
background concentration much of which represents anthropogenic releases accumulated
over the decades (UNEP, 2002).
Hg in the atmosphere exists in three major inorganic forms including gaseous elemental
mercury (GEM, $Hg^0$), gaseous oxidized mercury (GOM, $Hg^{2+}$) and particulate bound
mercury (PBM, Hg(p)). GEM which is the dominant form of Hg in ambient air, (>95%) has a
relatively long residence time (0.5~2 years) due to its low reactivity and solubility (Schroeder
and Munthe, 1998). However, GOM has high water solubility and relatively strong surface
adhesion properties (Han et al., 2005), so it has a short atmospheric residence time (~days).
PBM is associated with airborne particles such as dust, soot, sea-salt aerosols, and ice crystals
(Lu and Schroeder, 2004) and is likely produced, in part, by adsorption of GOM species such
as $HgCl_2$ onto atmospheric particles (Gauchard et al., 2005; Lu and Schroeder, 2004; Sakata
and Marumoto, 2005; Seo et al., 2012; Seo et al., 2015).
Atmospheric Hg released from natural (e.g., volcanoes, volatilization from aquatic and
terrestrial environments) (Pirrone et al., 2010; Strode et al., 2007) and anthropogenic sources
(e.g., coal combustion, cement production, ferrous and non-ferrous metals manufacturing
facilities, waste incineration and industrial boilers) (Pacyna et al., 2010; Pacyna et al., 2006;
Pacyna et al., 2003; Pirrone et al., 2010; Zhang et al., 2015) when introduced into terrestrial
and aquatic ecosystem through wet and dry deposition (Mason and Sheu, 2002) can undergo
various physical and chemical transformations before being deposited. Its lifetime in the
atmosphere depends on its reactivity and solubility so that, depending on its form, it can have
impacts on local, regional and global scales (Lin and Pehkonen, 1999; Lindberg et al., 2007).
A portion of the Hg deposited in terrestrial environments through direct industrial discharge
or atmospheric deposition is transported to aquatic system through groundwater and surface
water runoff (Miller et al., 2013). A previous study also reported that Hg directly released
into terrestrial and aquatic ecosystems from industrial effluent has influenced surface water,
sediment and biological tissue (Flanders et al., 2010). Significant spatial variations in
atmospheric Hg deposition near urban and industrial areas are due to local anthropogenic
sources including municipal waste incinerators, medical waste incinerators, electric power
generating facilities and cement kilns (Dvonch et al., 1998), ferrous and non-ferrous metal
processing, iron and steel manufacturing facilities, oil and coal combustion (Hoyer et al.,
1995), and other forms of industrial combustion (Brown et al., 2015). Miller et al. (2013) also
reported that local sources of elemental Hg are typically industrial processes including retort
facilities used in the mercury mining industry to convert Hg containing minerals to elemental
Hg and chlor-alkali facilities.
The annual average national anthropogenic Hg emissions from South Korea in 2007 have
been estimated to be 12.8 tons (range 6.5 to 20.2 tons); the major emission sources are coal
combustion in thermal power plants (25.8%), oil refineries (25.5%), cement kilns (21%),
incinerators (19.3%) including sludge incinerators (4.7%), municipal waste incinerators
(MWIs) (3%), industrial waste incinerators (IWIs) (2.7%), hospital/medical/infectious waste
incinerators (HMIWIs) (8.8%), and iron manufacturing (7%) (Kim et al., 2010). Global
anthropogenic Hg emissions were estimated to be 1960 tons in 2010 with East and Southeast
Asia responsible for 777 tons (39.7%) (19.6 tons for Japan and 8.0 tons for South Korea)
(AMAP/UNEP, 2013). China is the largest Hg emitting country in the world, contributing
more than 800 tons (~ 40%) of the total anthropogenic Hg emissions (UNEP, 2008).
Background atmospheric Hg concentrations in the northern hemisphere have decreased
since 1996 (Slemr et al., 2003), as measured at the Global Atmosphere Watch (GAW) station
at Mace Head, Ireland (Ebinghaus et al., 2011) and at the Canadian Atmospheric Mercury
Network (CAMNet) (Temme et al., 2007). In urban areas in South Korea atmospheric TGM
concentrations have also decreased over the last few decades due to the reduced fossil fuel
(mainly anthracite coal) consumption (Kim et al., 2016; Kim and Kim, 2000). However, this
decreasing trend is inconsistent with steady or increasing global anthropogenic Hg emissions
since 1990 in the northern hemisphere (Streets et al., 2011; Weigelt et al., 2015; Wilson et al.,
2010). A previous study reported that the global anthropogenic Hg emissions are increasing
with an average of 1.3% annual growth without including the artisanal and small-scale
production sector (Muntean et al., 2014).
Receptor models are often used to identify sources of air pollutants and are focused on the
pollutants behavior in the ambient environment at the point of impact (Hopke, 2003). In
previous studies, conditional probability function (CPF), which utilizes the local wind
direction, and potential source contribution function (PSCF), which utilizes longer backward
trajectories (typically 3-5 days), combined with concentration data were used to identify
possible transport pathways and source locations (Hopke, 2003). While PSCF has been used
primarily to identify regional sources, it has also been used to identify local sources (Hsu et
al., 2003).
The objectives of this study were to characterize the hourly and seasonal variations of
atmospheric TGM (the sum of the GEM and the GOM) concentrations, to identify the
relationships between TGM and co-pollutant concentrations, and to identify likely source
directions and locations of TGM using CPF, conditional bivariate probability function
(CBPF) and total PSCF (TPSCF).

**2.  Materials and methods**
*2.1. Sampling and analysis*

TGM concentrations were measured on the roof of the Korean Federation of

Community Credit Cooperatives (KFCCC) building (latitude: 35.992°, longitude: 129.404°,
~10 m above ground) in Pohang city, in Gyeongsangbuk-do, a province in eastern South
Korea. Gyeongsangbuk-do has a population of 2.7 million (5% of the total population and the
third most populated province in South Korea) and an area of 19,030 km$^2$ (19% of the total
area of South Korea and the largest province geographically in South Korea). Pohang city has
a population of 500,000 (1% of the total population in South Korea) and an area of 605.4 km$^2$
(1.1% of the total area in South Korea). It is heavily industrialized with the third largest steel
manufacturing facility in Asia and the fifth largest in the world. There are several iron and
steel manufacturing facilities including electric and sintering furnaces using coking in
Gyeongsangbuk-do including Pohang. In addition, there are several coke plants around the
sampling site. The Hyungsan River divides the city into a residential area and the steel
complex. Hg emissions data from iron and steel manufacturing, and a hazardous waste
incinerator were estimated based on a previous study (Kim et al., 2010) (Fig. 1).
TGM concentrations were measured every 5 min during summer (17 August~23 August
2012), fall (9 October~17 October 2012), winter (22 January ~29 January 2013), and spring
(26 March~3 April 2013) using a mercury vapor analyzer (Tekran 2537B) which has two
gold cartridges that alternately collect and thermally desorb mercury. Ambient air at a flow
rate of 1.5 L min$^{-1}$ was transported through a 3 m-long heated sampling line (1/4" OD Teflon)
in to the analyzer. The sampling line was heated at about 50 ℃ using heat tape to prevent
water condensation in the gold traps because moisture on gold surfaces interferes with the
amalgamation of Hg (Keeler and Barres, 1999). Particulate matter was removed from the
sampling line by a 47 mm Teflon filter.

*2.2. Meteorological data*
Hourly meteorological data (air temperature, relative humidity, and wind speed and
direction) were obtained from the Automatic Weather Station (AWS) operated by the Korea
Meteorological Administration (KMA) (http://www.kma.go.kr) (6 km from the site). Hourly
concentrations of $NO_2$, $O_3$, CO, $PM_{10}$ and $SO_2$ were obtained from the National Air Quality
Monitoring Network (NAQMN) (3 km from the site) (Fig. 1).
Meteorological Setting. Fig. S1 shows the frequency of counts of measured wind direction
occurrence by season during the sampling period. The predominant wind direction at the
sampling site was W (20.9%) and WS (19.2%), and calm conditions of wind speed less than
1 m s$^{-1}$ occurred 7.6% of the time. Compared to other seasons, however, the prevailing winds
in summer were N (17.0%), NE (16.4%), S (16.4%), and SW (15.8%).

*2.3. QA/QC*
Automated daily calibrations were carried out for the Tekran 2537B using an internal
permeation source. Two-point calibrations (zero and span) were separately performed for
each gold cartridge. Manual injections were performed prior to every field sampling
campaign to evaluate these automated calibrations using a saturated mercury vapor standard.
The relative percent difference (RPD) between automated calibrations and manual injections
was less than 2%. The recovery measured by directly injecting known amounts of four
mercury vapor standards when the sample line was connected to zero air ranged from 92 to
110% (99.4 ± 5.2% in average).

**3.  Model descriptions**
*3.1. Conditional Probability Function (CPF)*
CPF was originally performed to determine which wind directions dominate during high
concentration events to evaluate local source impacts (Ashbaugh et al., 1985). It has been
successfully used in many previous studies (Begum et al., 2004; Kim et al., 2003a; Kim et al.,
2003b; Xie and Berkowitz, 2006; Zhao et al., 2004; Zhou et al., 2004). CPF estimates the
probability that the measured concentration will exceed the threshold criterion for a given
wind direction. The CPF is defined as follows Eq. (1).
$$CPF_{\Delta\theta} = \frac{m_{\Delta\theta|c\geq x}}{n_{\Delta\theta}} \tag{1}$$

where, $m_{\Delta\theta}$ is the number of samples from the wind sector $\theta$ having concentration C greater
than or equal to a threshold value $x$, and $n_{\Delta\theta}$ is the total number of samples from wind sector
$\Delta\theta$. In this study, 16 sectors ($\Delta\theta = 22.5º$) were used and calm winds ($\leq 1$ m s$^{-1}$) were excluded
from the analysis. The threshold criterion was set at above the overall average TGM
concentration (5.0 ng m$^{-3}$). Thus, CPF indicates the potential for winds from a specific
direction to contribute to high air pollution concentrations.

*3.2. Conditional Bivariate Probability Function (CBPF)*
CBPF couples ordinary CPF with wind speed as a third variable, allocating the measured
concentration of pollutant to cells defined by ranges of wind direction and wind speed rather
than to only wind direction sectors.
The CBPF is defined as follows Eq. (2).

$$CBPF_{\Delta\theta,\Delta u} = \frac{m_{\Delta\theta,\Delta u}|_{C \geq x}}{n_{\Delta\theta,\Delta u}}$$
    (2)


where, $m_{\Delta\theta,\Delta u}$ is the number of samples in the wind sector $\Delta\theta$ with wind speed interval $\Delta u$
having concentration *C* greater than a threshold value *x*, and $n_{\Delta\theta\Delta u}$ is the total number of
samples in that wind direction-speed interval. The threshold criterion was set at above the
overall average TGM concentration (5.0 ng m$^{-3}$). The extension to the bivariate case can
provide more information on the nature of the sources because different source types such as
stack emission sources and ground-level sources can have different wind speed dependencies
(prominent at high and low wind speed, respectively). More detailed information is described
in a previous study (Uria-Tellaetxe and Carslaw, 2014).

*3.3. Potential Source Contribution Function (PSCF)*
The PSCF model has been extensively and successfully used in the previous studies to
identify the likely source areas (Cheng et al., 1993; Han et al., 2004; Hopke et al., 2005; Lai
et al., 2007; Lim et al., 2001; Poissant, 1999; Zeng and Hopke, 1989). The PSCF is a simple
method that links residence time in upwind areas with high concentrations through a
conditional probability field and was originally developed by Ashbaugh et al. (1985). PSCF$_{ij}$
is the conditional probability that an air parcel that passed through the $ij$th cell had a high
concentration upon arrival at the monitoring site and is defined as the following Eq. (3).

$$PSCF_{ij} = \frac{m_{ij}}{n_{ij}} \tag{3}$$

where, $n_{ij}$ is the number of trajectory segment endpoints that fall into the $ij$-th cell, and $m_{ij}$ is the
number of segment endpoints in the same grid cell ($ij$-th cell) when the concentrations are higher
than a criterion value as measured at the sampling site.
High PSCF values in those grid cells are regarded as possible source locations. Cells including
emission sources can be identified with conditional probabilities close to one if trajectories that
have crossed the cells efficiently transport the released pollutant to the receptor site. Therefore,
the PSCF model provides a tool to map the source potentials of geographical areas.
The criterion value of PSCF for TGM concentration was set at above the overall average
concentration (5.0 ng m$^{-3}$) to identify the emission sources associated with high TGM
concentrations and provide a better estimation and resolution of source locations during the
sampling periods. The geographic area covered by the computed trajectories was divided into
an array of 0.05° latitude by 0.05° longitude grid cells. As will be discussed in Section 5.3, 24
h backward trajectories starting at every hour at a height of 10, 50, and 100 m above ground
level were computed using the vertical velocity model because local sources are more
important than that of long-range transport in this study (It should be noted that PSCF results
using 48 h backward trajectories had similar results as the 24 h backward trajectories). Each
trajectory was terminated if they exit the model top (5,000m), but advection continues along
the surface if trajectories intersect the ground. To generate horizontally highly resolved
meteorological inputs for trajectory calculations, the Weather Research and Forecast (WRF)
model was used to generate a coarse domain at a resolution of 27 km and a nested domain at
a horizontal resolution of 9 km, which geographically covers northeast Asia and the southern
part of the Korean Peninsula, respectively. The nested domain has 174 columns in the east-
west direction and 114 rows in the north-south direction. PSCF was calculated with 9 km
meteorological data.

In this study, TPSCF which incorporates probability from above different starting

heights was calculated since backward trajectories starting at different heights traverse
different distances and pathways, thus providing information that cannot be obtained from a
single starting height (Cheng et al., 1993).

Previous studies suggest that there are increasing uncertainties as backward trajectory

distances increase (Stohl et al., 2002) and that PSCF modeling is prone to the trailing effect is
which locations upwind of sources are also identified as potential sources (Han et al., 2004).
An alternative to back trajectory calculations in the interpretation of atmospheric trace
substance measurements (Stohl et al., 2002) although this technique does not provide much
information on source locations.

Generally, PSCF results show that the potential sources covered wide areas instead of

indicating individual sources due to the trailing effect. The trailing effect appears since PSCF
distributes a constant weight along the path of the trajectories. To minimize the effect of
small $n_{ij}$ (the number of trajectory segment endpoints that fall into the $ij$-th cell) values,
resulting in high TPSCF values with high uncertainties, an arbitrary weight function $W(n_{ij})$
was applied to down-weight the PSCF values for the cell in which the total number of end
points was less than three times the average value of the end points (Choi et al., 2011; Heo et
al., 2009; Hopke et al., 1995; Polissar et al., 2001). The TPSCF value for a grid cell was
defined with following Eq. (4).

$$P(TPSCF_{ij}) = \frac{P(m_{ij})_{10m} + P(m_{ij})_{50m} + P(m_{ij})_{100m}}{P(n_{ij})_{10m} + P(n_{ij})_{50m} + P(n_{ij})_{100m}} \times W \qquad (4)$$


where,

$$W(n_{ij}) = \begin{cases} 1.0, & 3n_{ave} < n_{ij} \\ 0.8, & 2n_{ave} < n_{ij} \leq 3n_{ave} \\ 0.6, & n_{ave} < n_{ij} \leq 2n_{ave} \\ 0.4, & 0.5n_{ave} < n_{ij} \leq n_{ave} \\ 0.2, & n_{ij} \leq 0.5n_{ave} \end{cases}$$


**4.  Clean Air Policy Support System (CAPSS) data**
In this study, the Korean National Emission Inventory estimated using Clean Air Policy
Support System (CAPSS) data developed by the National Institute of Environmental
Research (NIER) were used (http://airemiss.nier.go.kr/main.jsp (accessed December 09,
2015)). The CAPSS is the national emission inventory system for the air pollutants (CO,
NOx, SOx, TSP, PM$_{10}$, PM$_{2.5}$, VOCs and NH$_3$) which utilizes various national, regional and
local statistical data collected from about 150 organizations in Korea. In CAPSS, the Source
Classification Category (SCC) excluding fugitive dust and biomass burning based on the
European Environment Agency's (EEA) CORe Inventory of AIR emissions was classified
into the following four levels (EMEP/CORINAIR) (NIER, 2011).
(1) The upper level (SCC1): 11 source categories ,
(2) The intermediate level (SCC2): 42 source categories and
(3) The lower level (SCC3): 173 source categories

The sectoral contributions of emissions of South Korea, Gyeongsangbuk-do and Pohang
for CO, NOx, SOx, TSP, $PM_{10}$, $PM_{2.5}$, VOC and $NH_3$ are shown in Fig. S2 (See SI for
details).
More detailed information about SCCs in CAPSS is described in Table S1.

**5.   Results and Discussions**
*5.1. General characteristics of TGM*
The seasonal distributions of TGM were characterized by large variability during each
sampling period (Fig. 2). The average concentration of TGM during the complete sampling
period was $5.0 \pm 4.7$ ng m$^{-3}$ (range: 1.0-79.6 ng m$^{-3}$). This is significantly higher than the
Northern Hemisphere background concentration (~1.5 ng m$^{-3}$) (Sprovieri et al., 2010) and
those measured in China, in Japan and other locations in Korea, however lower than those
measured at Changchun, Gui Yang and Nanjing in China (Table 1). The median TGM
concentration was 3.6 ng m$^{-3}$ which was much lower than that of the average, suggesting that
there were some extreme pollution episodes with very high TGM concentrations.
The TGM concentration follows a typical log-normal distribution (Fig. S3). The range of 2
to 5 ng m$^{-3}$ dominated the distribution, accounting for more than half of the total number of
samples (60.8%). The maximum frequency of 28.1% occurred between 2 and 3 ng m$^{-3}$.
Extremely high TGM concentration events (>20 ng m$^{-3}$) were also observed (1.7% of the
time).



*5.2. Seasonal variations*
The TGM concentration was statistically significantly higher in fall ($6.7 \pm 6.4$ ng m$^{-3}$) ($p <$
0.01), followed by spring ($4.8 \pm 4.0$ ng m$^{-3}$), winter ($4.5 \pm 3.2$ ng m$^{-3}$) and summer ($3.8 \pm 3.9$
ng m$^{-3}$) (Table 2). The highest concentrations (TGM > 10 ng m$^{-3}$) were measured more
frequently in fall (24.7%), and the lowest concentrations (TGM < 3 ng m$^{-3}$) mainly occurred
in summer (49.7%). The low TGM concentration in summer is likely because increased
mixing height (Friedli et al., 2011), and gas phase oxidation (Choi et al., 2013; Huang et al.,
2010; Lynam and Keeler, 2006) at higher temperatures particularly at this sampling site
which is close to the ocean (2 km) where oxidation involving halogens may be enhanced
(Holmes et al., 2009; Lin et al., 2006). The high TGM concentrations in fall was due to
different wind direction (see Fig. S1), sources, relationships with other pollutants and
meteorological conditions. More detailed information can be found in Section 5.4.

The average concentrations of $NO_2$, $O_3$, CO, $PM_{10}$ and $SO_2$ during the complete sampling

period were $23.1 \pm 10.8$ ppbv, $24.6 \pm 12.5$ ppbv, $673.7 \pm 487.3$ ppbv, $55.5 \pm 26.4$ μg m$^{-3}$ and
$6.7 \pm 4.3$ ppbv, respectively. $NO_2$, $O_3$, CO, $PM_{10}$ and $SO_2$ concentrations were highest in
spring (Table 2). There was a statistically significant positive correlation between the TGM
and $PM_{10}$ (r = 0.10) ($p < 0.01$). However, the TGM concentration was not significantly
correlated with $NO_2$, CO or $SO_2$ concentrations, suggesting that combustion associated with
space heating was not a significant source of TGM (Choi et al., 2009).

*5.3. Relationship between TGM and CO*

CO has a significant anthropogenic source and is considered to be an indicator of

anthropogenic emissions (Mao et al., 2008). Previous studies reported that TGM and CO
have a strong correlation because they have similar emission sources (combustion processes)
and similar long atmospheric residence times (Weiss-Penzias et al., 2003).

There was a weak positive correlation between TGM and CO in this study (r = 0.04) ($p$ =

0.27). However there was a statistically significant correlation between TGM and CO in
winter (r = 0.25) ($p < 0.05$), suggesting that TGM and CO were affected by similar, possibly
distant, anthropogenic emission sources in winter.

On the other hand, there were no statistically significant correlations between TGM and

CO in spring (r = 0.02) (*$p = 0.78$*), in summer (r = 0.13) (*$p = 0.08$*), or in fall (r = -0.03) (*$p =$*
*0.69*), indicating that TGM and CO were affected by different anthropogenic emission
sources in these seasons.

Previous studies identified the long-range transport of mercury using the $\Delta$TGM/$\Delta$CO

enhancement ratio (Choi et al., 2009; Jaffe et al., 2005; Kim et al., 2009; Weiss-Penzias et al.,
2003; Weiss-Penzias et al., 2006). Kim et al. (2009) and Choi et al. (2009) investigated high
concentration events which were defined as at least a 10 h period with hourly average TGM
and CO concentrations higher than the average monthly TGM and CO concentrations. They
reported that long-range transport events were characterized by high values of TGM/CO ratio
($\Delta$TGM/$\Delta$CO) (0.0052-0.0158 ng m$^{-3}$ ppb$^{-1}$) and high correlations ($r^2 > 0.5$), whereas local
events showed low $\Delta$TGM/$\Delta$CO (0.0005 ng m$^{-3}$ ppb$^{-1}$ in average) and weak correlations ($r^2 <$

0.5).

The observed $\Delta$TGM/$\Delta$CO was 0.0001 ng m$^{-3}$ ppb$^{-1}$ in spring, 0.0005 ng m$^{-3}$ ppb$^{-1}$ in

summer, -0.0007 ng m$^{-3}$ ppb$^{-1}$ in fall, 0.0011 ng m$^{-3}$ ppb$^{-1}$ in winter, which are significantly
lower than that indicative of Asian long-range transport (0.0046-0.0056 ng m$^{-3}$ ppb$^{-1}$) (Friedli
et al., 2004; Jaffe et al., 2005; Weiss-Penzias et al., 2006), suggesting that local sources are
more important than that of long-range transport in this study. The  ΔTGM/ΔCO in winter
(0.0011 ng m$^{-3}$ ppb$^{-1}$) was similar to that of a site impacted by local sources in Korea (Kim et
al., 2009) and in US industrially related events (0.0011 ng m$^{-3}$ ppb$^{-1}$) (Weiss-Penzias et al.,

2007).

There are also uncertainties from the potential mixing between Hg associated with long-

range transported airflows and local air making it difficult to distinguish between distant and
local source impacts. However, it is possible that the one-week sampling period in each
season did not capture the long-range transport events, and more can be learned using a larger
dataset than just using the one-week sampling period to confirm these results.

*5.4. Diurnal variations*

Diurnal variations of TGM (Fig. 3), co-pollutants concentrations, and meteorological

data were observed (Fig. S4). TGM, $O_3$, CO, $SO_2$, and temperature in the daytime (06:00-
18:00) were higher than those in the nighttime (18:00-06:00) ($p < 0.05$) except $PM_{10}$ ($p =$
0.09) (Fig. S5). However, $NO_2$ during the nighttime because of relatively lower
photochemical reactivity with $O_3$ was higher than that in daytime ($p < 0.05$) (Adame et al.,

2012).

The daytime TGM concentration (5.3 ± 4.7 ng m$^{-3}$) was higher than that in the nighttime

(4.7 ± 4.7 ng m$^{-3}$) ($p < 0.01$), which was similar to several previous studies (Cheng et al.,
2014; Gabriel et al., 2005; Nakagawa, 1995; Stamenkovic et al., 2007) but different than
another studies (Lee et al., 1998). Previous studies reported that this different is due to local
sources close to the sampling site (Cheng et al., 2014; Gabriel et al., 2005), a positive
correlation between TGM concentration and ambient air temperature (Nakagawa, 1995) and
increased traffic (Stamenkovic et al., 2007). However, another study suggested that the higher
TGM concentration during the night was due to the shallowing of the boundary layer, which
concentrated the TGM near the surface (Lee et al., 1998).

In a previous study the daytime TGM concentration was relatively lower than that in the

nighttime because the sea breeze transported air containing low amounts of TGM from the
ocean during the daytime whereas the land breeze transported air containing relatively high
concentrations of TGM from an urban area during the nighttime (Kellerhals et al., 2003).
Although it is possible that the land-sea breeze may affect diurnal variations in TGM
concentrations since the sampling site was near the ocean and lower TGM were also observed
during the daytime, the higher concentrations in the daytime than those in nighttime were due
to local emission sources because the daytime temperature ($14.7 \pm 10.0$ ℃) was statistically
significantly higher than that in the nighttime ($13.0 \pm 9.8$ ℃) (t-test, $p < 0.05$) and there was a
weak but statistically significant negative correlation between TGM concentration and
ambient air temperature (r = -0.08) ($p < 0.05$). In addition, there are several known Hg
sources such as iron and steel manufacturing facilities including electric and sintering
furnaces using coking between the sampling site and the ocean.

As shown in Fig. 3 and Fig. S4, there was a weak but negative relationship between the

TGM concentrations and $O_3$ concentrations (r = -0.18) ($p < 0.01$), suggesting that oxidation
of GEM in the oxidizing atmosphere during periods of strong atmospheric mixing was
partially responsible for the diurnal variations of TGM concentrations. In addition, oxidation
of GEM by bromine species in the coastal area (Obrist et al., 2011) or by chloride radicals in
marine boundary layer (Laurier et al., 2003) might play a significant role. If oxidation of
GEM occurred, GOM concentrations would increase. However there are uncertainties on the
net effects on TGM (the sum of the GEM and the GOM) since we did not measure GOM
concentrations.

TGM concentration was negatively correlated with ambient air temperature *(r = -0.08)*

*(p < 0.05)* because high ambient air temperature in the daytime will increase the height of the
boundary layer and dilute the TGM, and the relatively lower boundary layer at nighttime
could concentrate the TGM in the atmosphere (Li et al., 2011). Although there was a
statistically significant negative correlation between the TGM concentration and ambient air
temperature, there was a rapid increase in TGM concentration between 06:00-09:00 when
ambient temperatures also increased possibly due to local emissions related to industrial
activities, increased traffic, and activation of local surface emission sources. Similar patterns
were found in previous studies (Li et al., 2011; Stamenkovic et al., 2007). Nonparametric
correlations revealed that there is a weak positive correlation between TGM and ambient air
temperature ($r_s = 0.11$, *p=0.27*) between 06:00-09:00. The TGM concentration was negatively
correlated with $O_3$ ($r_s = -0.33$, *p<0.01*) but positively correlated with $NO_2$ ($r_s = 0.21$, *p<0.05*),
suggesting that the increased traffic is the main source of TGM during these time periods.
Compared to other seasons, significantly different diurnal variations of TGM were
observed in fall. The daytime TGM concentrations in fall were similar to those in other
seasons, however, the nighttime TGM concentrations in fall were much higher than other
seasons. As described earlier in Section 5.2, the high TGM concentrations in fall was
possibly due to the relationship between other pollutants and meteorological conditions as
well as different wind direction and sources. The nighttime TGM concentrations in fall were
simultaneously positively correlated with $PM_{10}$ (r=0.26) (*p<0.05*) and CO (r=0.21) (*p<0.05*)
concentrations and wind speed (r=0.35) (*p<0.01*), suggesting that the combustion process is
an important source during this period.
TGM generally showed a consistent increase in the early morning (06:00-09:00) and a
decrease in the afternoon (14:00-17:00), similar to previous studies (Dommergue et al., 2002;
Friedli et al., 2011; Li et al., 2011; Liu et al., 2011; Mao et al., 2008; Shon et al., 2005; Song
et al., 2009; Stamenkovic et al., 2007). Significantly different diurnal patterns have been
observed at many suburban sites with the daily maximum occurring in the afternoon (12:00-
15:00), possibly due to local emission sources and transport (Fu et al., 2010; Fu et al., 2008;
Kuo et al., 2006; Wan et al., 2009). Other studies in Europe reported that TGM
concentrations were relatively higher early in the morning or at night possibly due to mercury
emissions from surface sources that accumulated in the nocturnal inversion layer (Lee et al.,
1998; Schmolke et al., 1999).
Based on the above results, the diurnal variations in TGM concentration are due to a
combination of: 1) reactions with an oxidizing atmosphere, 2) changes in ambient
temperature and 3) local emissions related to industrial activities. To supplement these
conclusions CPF and CBPF were used to identify source directions and TPSCF was used to
identify potential source locations.

*5.5. CPF, CBPF and TPSCF results of TGM*
Conventional CPF, CBPF and TPSCF plots for TGM concentrations higher than the
average concentration show high source probabilities to the west in the direction of large steel
manufacturing facilities and waste incinerators (Fig. 4). The CPF only shows high
probabilities from the west and provides no further information, however, the CBPF shows
groups of sources with the high probabilities from the west and the northeast. CBPF shows
that the high probabilities from the west occurred under high wind speed ($> 3$ m s$^{-1}$)
indicative of emissions from stacks as well as low wind speed ($\leq 3$ m s$^{-1}$) indicative of non-
buoyant ground level sources (Uria-Tellaetxe and Carslaw, 2014).
As described in Section 5.3, correlations between TGM and CO revealed that TGM and
CO were affected by similar anthropogenic emission sources in winter but affected by
different sources in spring, summer and fall, which is supported by Fig. S6 which shows
significantly different seasonal patterns of CPF and CBPF for TGM concentrations.
However, compared to Fig. 4, the CPF and CBPF patterns in fall were similar to those during
the whole sampling periods. Especially, the nighttime TGM concentration in fall was
simultaneously positively correlated with PM$_{10}$ (r=0.26) ($p<0.05$) and CO (r=0.21) ($p<0.05$)
concentrations and wind speed (r=0.35) (p<0.01), indicating that the combustion process
from the west is an important source during this period.
Since TGM showed a significant correlation with CO (r=0.25) ($p<0.05$) and showed a
weak positive correlation with PM$_{10}$ (r=0.08) ($p=0.33$) in winter with high wind speed,
combustion sources from the west are likely partially responsible for this result.
TPSCF identified the likely sources of TGM as the iron and manufacturing facilities and
the hazardous waste incinerators which are located to the west from the sampling site. A
previous study reported that the waste incinerators (9%) and iron and steel manufacturing
(7%) were relatively high Hg emissions sources in Korea (Kim et al., 2010). Waste
incinerators emissions were due to the high Hg content in the waste (Lee et al., 2004).
Emissions from iron and steel manufacturing are due to the numerous electric and sintering
furnaces using coking which emits relatively high mercury concentrations (Lee et al., 2004)
in Gyeongsangbuk-do including Pohang. There are several coke plants around the sampling
site (http://www.poscoenc.com/upload/W/BUSINESS/PDF/ENG_PLANT_2_1_3_5.pdf
(accessed December 09, 2015)). They are essential parts of the iron and steel manufacturing,
and the major source of atmospheric mercury related to the iron and steel manufacturing is
from coke production (Pacyna et al., 2006).

The coastal areas east of the sampling site where there are large ports were also identified

as the likely source areas of TGM. A previous study reported that the emissions of gaseous
and particulate pollutants were high during vehicular operations in port areas and from
marine vessel and launches (Gupta et al., 2002). Another possibility is that significant amount
of GEM are emitted from the ocean surface because of photo-chemically and
microbiologically mediated photo-reduction of dissolved GOM (Amyot et al., 1994; Zhang
and Lindberg, 2001). The northeast direction including the East Sea was also identified as
potential source areas likely because this is an area with lots of domestic passenger ships
routes. The south from the sampling site was also identified as a likely source area of TGM
where Ulsan Metropolitan City, South Korea's seventh largest metropolis with a population
of over 1.1 million is located. It includes a large petrochemical complex known as a TGM
source (Jen et al., 2013).

**Conclusions**
During the sampling periods, the average TGM concentration was higher than the Northern
Hemisphere background concentration, however, considerably lower than those near urban
areas in China and higher than those in Japan and other locations in Korea. The median
concentration of TGM was much lower than that of the average, suggesting that there were
some extreme pollution episodes with very high TGM concentrations. The TGM
concentration was highest in fall, followed by spring, winter and summer. The high TGM
concentration in fall is due to transport from different wind directions than during the other
periods. The low TGM concentration in summer is likely due to increased mixing height and
gas phase oxidation at higher temperatures particularly at this sampling site which is close to
the ocean (2 km) where oxidation involving halogens may be enhanced.
TGM consistently showed a diurnal variation with a maximum in the early morning
(06:00-09:00) and minimum in the afternoon (14:00-17:00). Although there was a statistically
significant negative correlation between the TGM concentration and ambient air temperature,
the daytime TGM concentration was higher than those in the nighttime, suggesting that local
emission sources are important. There was a negative relationship between the TGM
concentrations and $O_3$ concentrations, indicating that the oxidation was partially responsible
for the diurnal variations of TGM concentrations. The observed $\Delta TGM/\Delta CO$ was
significantly lower than that indicative of Asian long-range transport, suggesting that local
sources are more important than that of long-range transport. CPF only shows high
probabilities to the west from the sampling site where there are large steel manufacturing
facilities and waste incinerators. However, CBPF and TPSCF indicated that the dominant
sources of TGM were the hazardous waste incinerators and the coastal areas in the northeast
as well as the iron and manufacturing facilities in the west. The domestic passenger ships
routes in the East Sea were also identified as possible source areas.

**Author contribution**
Yong-Seok Seo conducted a design of the study, the experiments and analysis of data, wrote
the initial manuscript, and finally approved the final manuscript. Seung-Pyo Jeong, Eun Ha
Park, Tae Young Kim, Hee-Sang Eum, Dae Gun Park, Eunhye Kim, Jaewon Choi and Jeong-
Hun Kim conducted the experiments, analysis of data, and finally approved the final
manuscript. Thomas M. Holsen, Young-Ji Han and Eunhwa Choi and Soontae Kim
conducted interpretation of the results, revision of the initial manuscript, and finally approved
the final manuscript. Seung-Muk Yi conducted a design of the study, acquisition of data of the
study, interpretation of data, and revision of the initial manuscript, and finally approved the final
manuscript.

**Acknowledgments**
We thank National Institute of Environmental Research (NIER) for providing CAPSS data.
This work was supported by the National Research Foundation of Korea (NRF) (NRF-2008-
0059001), the Korean Ministry of Environment (MOE) as "the Environmental Health Action
Program (2015001370001) and the Brain Korea 21 (BK21) Plus Project (Center for Healthy
Environment Education and Research) through the National Research Foundation (NRF).

**Table List**
Table 1. Comparison with previous studies for TGM concentrations.
Table 2. Summary of atmospheric concentrations of TGM and co-pollutants, and
meteorological data.

**Figure List**
Fig. 1. The location of sampling site in this study ((a) South Korea, (b) Gyeongsangbuk-do
and (c) Pohang).
Fig. 2. Time-series of TGM concentrations in this study.
Fig. 3. The diurnal variations of TGM concentrations during the sampling periods.
Fig. 4. CPF, CBPF and TPSCF plots for TGM higher than average concentration.
**Table 1.** Comparison with previous studies for TGM concentrations.

| Country | Location | Sampling period | TGM conc. (ng m$^{-3}$) | Classifications | Reference |
|---------|----------|-----------------|-------------------------|-----------------|-----------|
| China | Mt. Hengduan, Qinghai–Tibet Plateau | Jul. 2010 ~ Oct. 2010 | 2.5 | Remote | Fu et al. (2015) |
| China | Nanjing, Jiangsu | Jan. 2011 ~ Oct. 2011 | 7.9 | Urban | Hall et al. (2014) |
| China | Mt. Dinghu, Guangdong | Oct. 2009 ~ Apr. 2010 | 5.1 | Rural | Chen et al. (2013) |
| China | Guangzhou, Guangdong | Nov. 2010 ~ Nov. 2011 | 4.6 | Urban | Chen et al. (2013) |
| China | Gui Yang, Guizhou | Jan. 2010 ~ Feb. 2010 | 8.4 | Urban | Feng et al. (2004) |
| China | Changchun, Jilin | Jul. 1999 ~ Jul. 2000 | 13.5-25.4 | Urban | Fang et al. (2004) |
| Japan | Fukuoka | Jun. 2012 ~ May 2013 | 2.33 | Urban | Marumoto et al. (2015) |
| Japan | Tokai-mura | Oct. 2005 ~ Aug. 2006 | 3.8 | Suburban | Osawa et al. (2007) |
| Japan | Tokyo | Apr. 2000 ~ Mar. 2001 | 2.7 | Urban | Sakata and Marumoto (2002) |
| Korea | Seoul | 1987 ~ 2013 | 3.7 | Urban | Kim et al. (2016) |
| Korea | Gangwon-do, Chuncheon | 2006 ~ 2009 | 2.1 | Rural | Han et al. (2014) |
| Korea | Seoul | Feb. 2005 ~ Feb. 2006 | 3.2 | Urban | Kim et al. (2009) |
| Korea | Seoul | Feb. 2005 ~ Dec. 2006 | 3.4 | Urban | Choi et al. (2009) |
| Korea | Seoul | 19 Sep. 1997 ~ 29 Sep. 1997<br>27 May. 1998 ~ 18 Jun. 1998 | 3.6 | Urban | Kim and Kim (2001) |
| Korea | Gyeongsangbuk-do, Pohang | 17 Aug. 2012 ~ 23 Aug. 2012<br>9 Oct. 2012 ~ 17 Oct. 2012<br>22 Jan. 2013 ~ 29 Jan. 2013<br>26 Mar. 2013 ~ 3 Apr. 2013 | 5.0 | Urban | This study |


**Table 2.** Summary of atmospheric concentrations of TGM and co-pollutants, and meteorological data. Note that TGM was measured every 5-
min, and other pollutants and meteorological data were measured every 1-hour.

| | | TGM (ng m$^{-3}$) | NO$_2$ (ppb) | O$_3$ (ppb) | CO (ppb) | PM$_{10}$ (μg m$^{-3}$) | SO$_2$ (ppb) | Temperature (℃) | Wind speed (m s$^{-1}$) | Humidity (%) | Solar radiation (MJ m$^{-2}$) |
|---|---|---|---|---|---|---|---|---|---|---|---|
| **Spring** | N | 2139 | 189 | 215 | 215 | 215 | 215 | 216 | 216 | 216 | 216 |
| | Average | 4.8 ± 4.0 | 25.3 ± 9.0 | 29.4 ± 14.2 | 766.5 ± 505.2 | 70.1 ± 26.0 | 7.6 ± 3.8 | 10.5 ± 4.2 | 2.2 ± 1.2 | 56.2 ± 16.8 | 0.82 ± 1.09 |
| | Range | 1.9 – 45.3 | 8 – 55 | 2 – 58 | 300 – 3100 | 28 - 204 | 5 - 35 | 1.1 – 21.6 | 0.4 – 6.2 | 19.0 – 94.0 | 0 – 3.44 |
| **Summer** | N | 1863 | 187 | 188 | 187 | 188 | 188 | 186 | 180 | 186 | 141 |
| | Average | 3.8 ± 3.9 | 18.3 ± 9.2 | 18.9 ± 10.1 | 697.3 ± 689.7 | 35.1 ± 15.8 | 6.5 ± 6.2 | 26.6 ± 4.2 | 2.2 ± 1.1 | 82.5 ± 13.9 | 0.40 ± 0.69 |
| | Range | 1.2 – 75.9 | 4 – 44 | 5 – 48 | 200 – 3300 | 12 – 87 | 2 - 27 | 19.7 – 34.1 | 0.1 – 6.4 | 43 - 98 | 0 – 2.92 |
| **Fall** | N | 2226 | 212 | 212 | 212 | 212 | 211 | 216 | 216 | 216 | 216 |
| | Average | 6.7 ± 6.4 | 25.0 ± 7.8 | 23.7 ± 13.1 | 662.7 ± 350.2 | 58.1 ± 17.8 | 5.3 ± 3.5 | 17.4 ± 3.2 | 2.1 ± 0.8 | 54.5 ± 14.7 | 0.62 ± 0.90 |
| | Range | 1.0 – 79.6 | 9 – 53 | 6 – 69 | 300 – 2900 | 20 - 145 | 3 - 39 | 11.7 – 25.2 | 0.5 – 4.5 | 12 - 79 | 0 – 2.90 |
| **Winter** | N | 1917 | 188 | 187 | 188 | 188 | 186 | 192 | 192 | 192 | 192 |
| | Average | 4.5 ± 3.2 | 23.5 ± 14.7 | 26.1 ± 8.7 | 556.4 ± 298.9 | 56.3 ± 30.5 | 7.4 ± 2.5 | 1.1 ± 4.3 | 2.8 ± 1.1 | 46.3 ± 24.5 | 0.43 ± 0.71 |
| | Range | 1.3 – 66.4 | 5 – 74 | 1 – 41 | 200 – 2400 | 18 – 161 | 5 – 24 | -0.65 – 10.1 | 0.5 – 6.0 | 11 - 90 | 0 – 2.34 |
| **Total** | N | 8145 | 776 | 802 | 802 | 803 | 800 | 810 | 804 | 810 | 765 |
| | Average | 5.0 ± 4.7 | 23.1 ± 10.8 | 24.6 ± 12.5 | 673.7 ± 487.3 | 55.5 ± 26.4 | 6.7 ± 4.3 | 13.8 ± 9.9 | 2.3 ± 1.1 | 59.4 ± 22.1 | 0.59 ± 0.90 |
| | Range | 1.0 – 79.6 | 4 – 74 | 1 – 69 | 200 – 3300 | 12 – 204 | 2 – 39 | -6.5 – 34.1 | 0.1 – 6.4 | 11 - 98 | 0 – 3.44 |


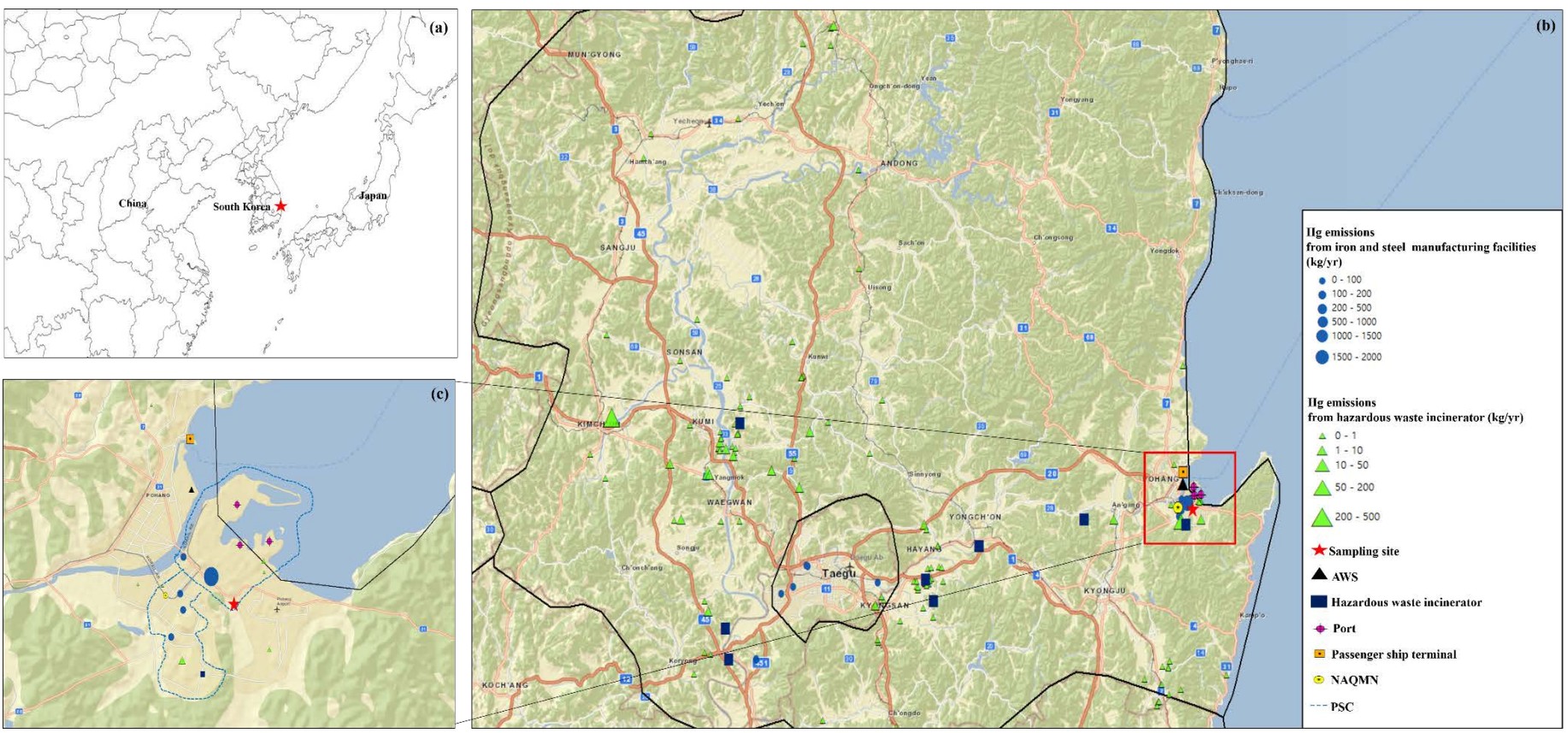

**Fig. 1.** The location of sampling site in this study ((a) South Korea, (b) Gyeongsangbuk-do and (c) Pohang). AWS, NAQMN and PSC represent Automatic Weather Station, National Air Quality Monitoring Network and Pohang Steel Complex, respectively.


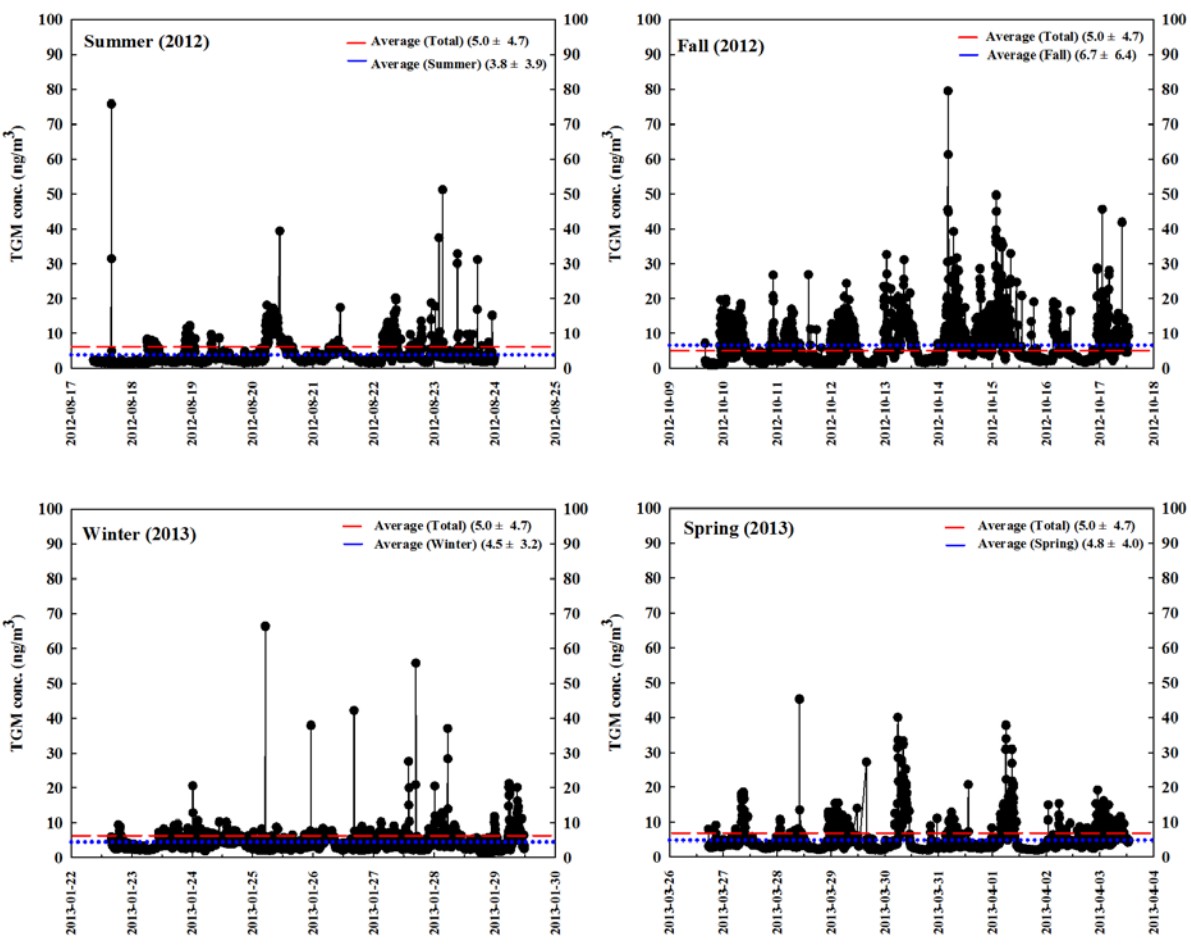


**Fig. 2.** Time-series of TGM concentrations in this study.

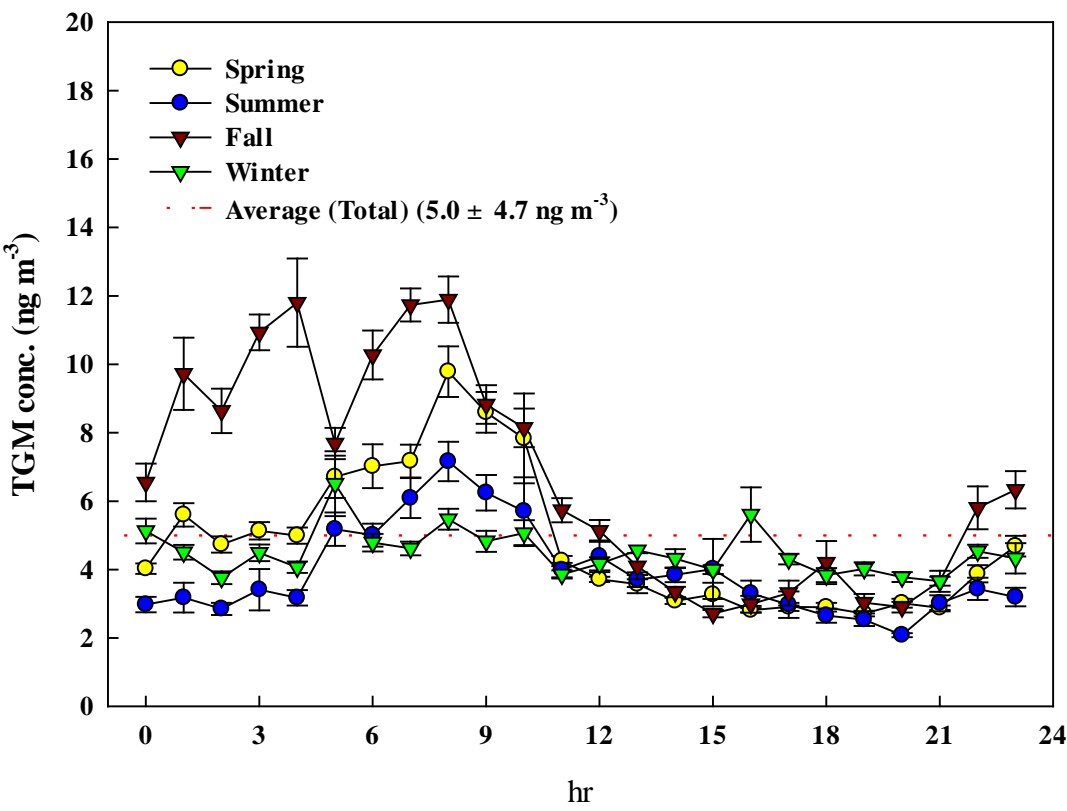

**Fig. 3.** The diurnal variations of TGM concentrations during the sampling periods.
The error bars represent standard error.

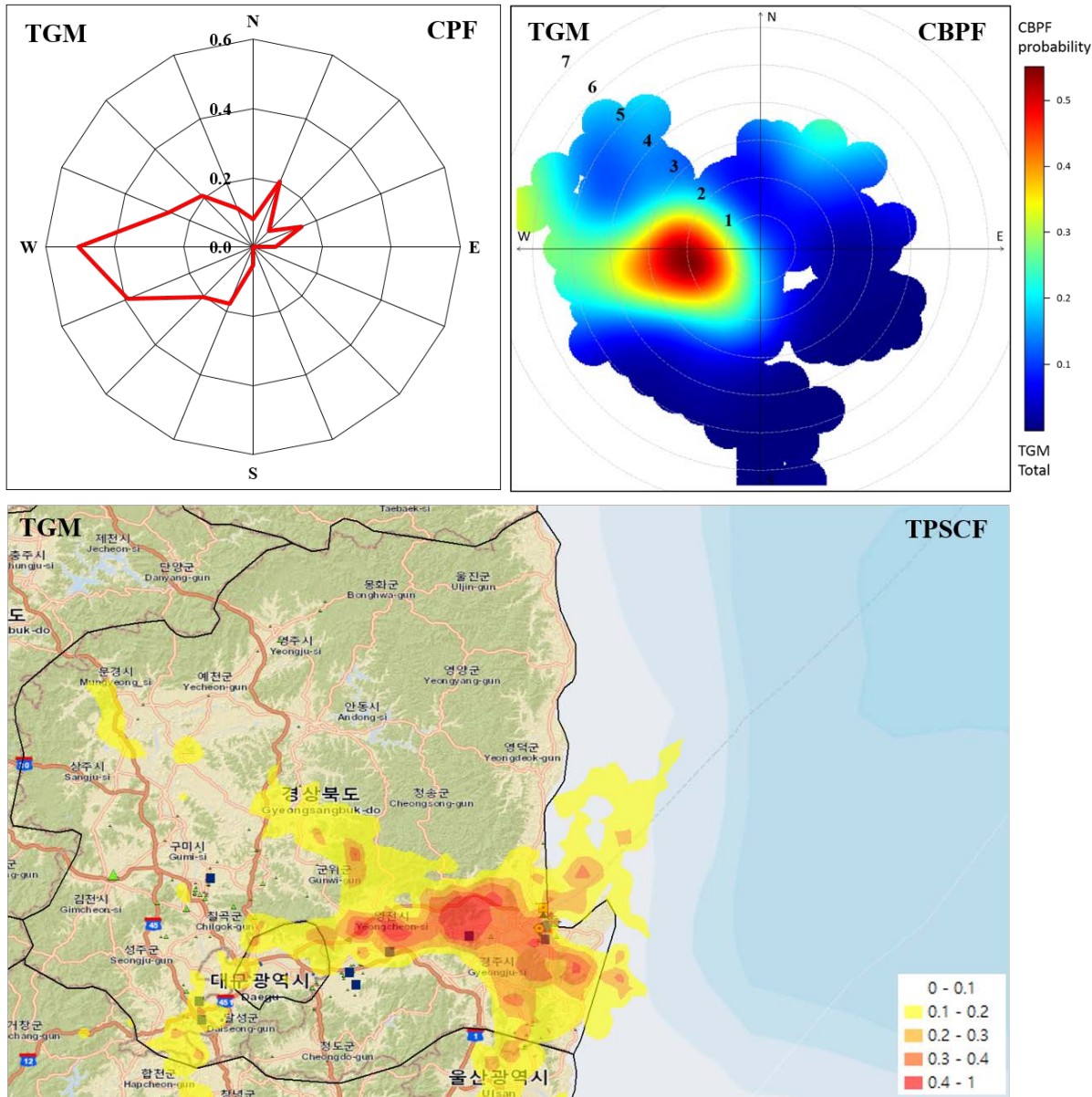

**Fig. 4.** CPF, CBPF and TPSCF plots for TGM higher than average concentration. The radial
axes of CPF and CBPF are the probability and the wind speed (m s$^{-1}$), respectively.

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

F. Updating Historical Global Inventories of Anthropogenic Mercury Emissions to
Air. AMAP Technical Report No. 3 (2010). Arctic Monitoring and Assessment
Programme (AMAP). 2010.
Xie, Y., and Berkowitz, C. M.: The use of positive matrix factorization with conditional
probability functions in air quality studies: an application to hydrocarbon emissions in
Houston, Texas, Atmos. Environ., 40, 3070-3091, 2006.
Zeng, Y., and Hopke, P.: A study of the sources of acid precipitation in Ontario, Canada,
Atmospheric Environment (1967), 23, 1499-1509, 1989.
Zhang, H., and Lindberg, S. E.: Sunlight and iron (III)-induced photochemical production of
dissolved gaseous mercury in freshwater, Environ. Sci. Technol., 35, 928-935, 2001.
Zhang, L., Wang, S., Wang, L., Wu, Y., Duan, L., Wu, Q., Wang, F., Yang, M., Yang, H.,
Hao, J., and Liu, X.: Updated emission inventories for speciated atmospheric mercury
from anthropogenic sources in China, Environ. Sci. Technol., 49, 3185-94, 2015.
Zhao, W., Hopke, P. K., and Karl, T.: Source identification of volatile organic compounds in
Houston, Texas, Environ. Sci. Technol., 38, 1338-1347, 2004.
Zhou, L., Kim, E., Hopke, P. K., Stanier, C. O., and Pandis, S.: Advanced factor analysis on
Pittsburgh particle size-distribution data special issue of aerosol science and
technology on findings from the Fine Particulate Matter Supersites Program, Aerosol
Science and Technology, 38, 118-132, 2004.