# Peer review of "Characteristics of total gaseous mercury (TGM) concentrations in an"

_Atmospheric Chemistry and Physics, 2015_

## Referee Comment (RC1) · Anonymous Referee #1 · 12 Feb 2016

General comments

This study analyzed seasonal and diurnal variations of TGM at a sampling site in southern Korea. Sources of TGM affecting the sampling site were investigated by correlating TGM with other pollutants and meteorological data and applying several source-receptor methods utilizing wind direction and back trajectory data. A newer method called the conditional bivariate probability function (CBPF) was used in this study to identify sources of TGM. However, I did not find this method very effective at differentiating between ground and stack emission sources. I find that there are many uncertainties in the CBPF results as well as in the back trajectory results that haven't been addressed in this paper. I have concerns about the methodology (insufficient
[Figure]

TGM data, selection of trajectory duration) and interpretation of correlation analysis results. A discussion of how the results vary in the different seasons was also lacking in many places of the paper, even though the results are shown in the figures. Overall, I find that a major revision of this paper is necessary.

Specific comments

Line 57 – the correlation coefficient of r = -0.08 is very small. It's more accurate to state there is little correlation between TGM and air temperature

Line 84 – "Atmospheric Hg released from natural and anthropogenic sources leading to enhanced deposition" Please clarify this statement.

Lines 87-89 – Is this sentence about Hg emissions to the atmosphere and the biogeo-chemical cycling of Hg or the direct release of Hg from industrial effluent?

Lines 94 – "coal combustion and waste incinerators" was already mentioned in this sentence.

Line 108 – use "data" instead of "information"

Line 138 – What temperature was the heated sampling line maintained at? Why is a heated sampling line necessary for sampling TGM?

2.3 QA/QC – the measurements were made for a one week period in each season. How often were the manual injections performed? Was there any maintenance activities performed prior to re-deployment of the instrument each time? These are important QA/QC procedures to mention because the instruments were offline for a long period of time.

Line 168 – This definition of CPF doesn't seem right because it is not exactly the source contribution. You can replace this sentence with the one in line 177.

Lines 173-175 – were the wind data measured every 5 min similar to TGM or was it averaged to the nearest hour?

Lines 183-185 –How does this method account for the full distribution of concentrations rather than concentrations exceeding a threshold? Based on Eq. 2, CBPF analyzes the subset of concentrations above a threshold as well. Another thing is how does this method account for sources with different dispersion characteristics? The equation is based on horizontal wind speeds, which is advection rather than dispersion.

Lines 193-195 – This explanation is not clear. Can you give some examples of mercury sources with different wind speed dependencies?

Lines 201-215 – this part needs to be rewritten by improving on the wording

Line 220 –Please justify the use of 24 hr trajectories. TGM is mainly GEM which has a longer residence time and capable of long range transport. This means a longer trajectory duration would be more suitable.

Line 229 – should be "meteorological data"

Lines 293-294 – This point was not discussed in later sections

Lines 299-302 – It should be stated more clearly that combustion was not a source of TGM because of a lack of correlation between TGM and the other combustion pollutant markers.

Lines 315 – Fig. S4 shows the CPF and CBPF plots in each season. Should this be discussed in section 5.5? It's not clear how these plots relate to the correlation results.

Lines 326-333 – It's surprising that long-range transport from China did not impact this site considering that it affected elevated Hg events in Seoul, Japan, and North America in previous studies. I would think long-range transport impacts a larger region including this sampling location. The TGM/CO slopes during elevated Hg events in Seoul were attributed to both long-range transport and local source impacts (Choi et al., 2009). Is it possible that the one-week sampling period in each season did not capture the long-range transport events? More data is needed to confirm these results. There are also uncertainties from the potential mixing between long-range transported airflows

and local air making it difficult to distinguish between distant and local source impacts.

Lines 345-346 – what are the time periods for daytime and nighttime concentrations? This sentence states daytime TGM were higher during daytime than nighttime. But in the previous sentence, the minimum TGM concentration occurs in the afternoon.

Lines 353-355 – You discussed about the land-sea/lake breeze effect on TGM diel patterns from another study in the previous sentence. Does this atmospheric process affect this particular site since it is near the ocean and lower TGM were also observed during daytime?

Lines 355-356 – The negative correlation between TGM and temperature is very small (as mentioned in the abstract, r = -0.08) despite a significant p-value. It's more accurate to state there is little relationship between TGM and air temperature.

Lines 357-360 – Similar to the above comment, the correlation between TGM and O3 is too small (r = -0.18) to suggest that it is indicative of GEM oxidation. It's more correct to state there is little relationship between TGM and O3. If GEM oxidation occurred, GOM concentrations would increase. There are some uncertainties on the net effect on TGM (GEM+GOM).

Lines 369-378 – The results here are inconsistent. If the small negative correlation between TGM and temperature indicates an increase in mixing height which leads to a decrease in TGM, how can it explain surface emissions in the morning which should increase with temperature? Is there a positive correlation between TGM and temperature in the morning?

Section 5.4 – This section is lacking discussion on seasonal differences in the diurnal variation. The fall diurnal pattern appears very different from those in other seasons in Fig. 3. Can you discuss why the TGM were much higher overnight in the fall but daytime concentrations were similar to those in other seasons? Why was there a large drop in TGM from 4:00 to 5:00 in the fall?
[Figure]

Lines 381-383 – These conclusions are not well-supported by the correlation analyses because the correlation coefficients were very small.

Line 390 – What are the potential Hg sources from the northeast direction?

Lines 391-393 – Please clarify this sentence. Are the higher wind speeds associated with stack emissions and lower wind speeds associated with surface emissions? There are several issues or uncertainties with this point. (1) Wind speeds were only measured at one height. How can you tell that the lower wind speeds are from lower elevation and vice-versa? What is the height of the wind measurements? (2) As mentioned in the diurnal variation section, boundary layer mixing occurs during the day. Is it possible to distinguish between ground emissions and stack emissions? (3) It seems only the west directions had both high and low wind speeds, while the east directions had only lower wind speeds (Fig. S1). Thus, CBPF doesn't seem useful when there is a lack of wind speed variation. It appears that it is by coincidence that both ground level and stack emissions were identified in the west direction because there happened to be a wind speed variation from this direction. Based on these points, the CBPF results don't seem to reveal more about TGM sources than CPF. More discussion is needed on the relationship between specific sources and wind speeds. Instead of wind speed, what other variables would be useful for source identification using CBPF?

Lines 394-395 – Are there industrial sources south of the sampling site? High probability areas are also identified in this direction in the TPSCF plot in Fig. 4.

Fig. 4 – the source areas seem confined to the industrial complex near the sampling site because of the short trajectory duration (24 hrs). Use of longer trajectories would help expand the source region and identify potential regional transport to the site. In addition to this uncertainty, other PSCF uncertainties should be discussed.

Section 5.5 – This section is lacking discussion on the CPF and CBPF results in different seasons. The seasonal plots are shown in Fig. S4, but they are not discussed in this section.

Lines 421-423 – This conclusion was not discussed in the results section. The wind direction frequency plots in different seasons are shown in Fig. S1 but the results were not discussed in the paper.

Lines 427-430 – p-value was significant but the correlation coefficient (magnitude of the relationship) is too small. I also don't understand the logic in these results. There should be a positive TGM and temperature correlation if daytime TGM concentrations were related to surface emissions.

Line 434 – JP-PSCF was not used anywhere else in the paper. Did you mean TPSCF?

---

## Referee Comment (RC2) · Anonymous Referee #2 · 15 Feb 2016

General Comments:

This study measured TGM concentrations in South Korea and analyzed seasonal and diurnal variations of TGM. They also used the statistics analysis to correlate TGM with other pollutants and meteorological data. They tried to identify the possible TGM sources using CPF, CBPF, and TPSCF models. They found that the nearby local sources are more significant than others. Over all, this paper meets the original contributions and contains unique TGM data nearby industrial areas. The authors performed the appropriate modeling approaches to identify possible mercury sources. However, the presentation quality doesn't meet the ACP's requirement. I suggest that the paper should be carefully revised and edited prior to publication on ACP.
[Figure]

Specific comments:

Introduction section contains too basic and out of dated information. I suggest adding some recent mercury inventory/modeling studies in East Asia. Result and discussion contains unnecessarily much literature review. More discussions are needed. The author used statistics analysis in many places. Please provide the type of analysis in this paper.

Here are details below.

Line 118 – 133: it should be combined into one paragraph.

Line 133: So what are their results in Kim et al. 2010? What did they find?

Line 146-151: it doesn't fit in material & method section. Please move to results & discussion section.

Line 156-157: need to explain how often manual injections were conducted.

Line 164: already used CPF in Line 163. Replace "Conditional Probability Function (CPF)" with "CPF"

Line 220 – 222 and line 230 – 233 are same.

Line 237: what is nij values here?

Line 272-275: Can the author provide recent TGM data from China and other country?

Line 285 and later: if the author mentioned p-value ($p < 0.01$ or $p < 0.05$), "statistically significantly" does not have to be addressed every time. Readers already know that the author performed statistical analysis.

Line 293: "as will be discussed later…." Can you indicate where and which section it was discussed?

This 5.4 section is for result and discussion. It includes too much literature review rather than discussion.

Line 346 – 348: can you explain what previous studies concluded about these diurnal variations? Needs more discussion.

Line 353: "as will be discussed later…." Indicate where and which section it was discussed? What is "this" mean here? Does "this" mean lower TGM in daytime?

Line 355 – 356 and line 369 – 370 are same. Please rephrase or rewrite.

Line 369 – 378: this paragraph is vague. Please clarify.

This 5.5 section also has too much literature review rather than discussion.

Line 381 – 385: this paragraph is the result from the section 5.4. Please move it to section 5.4.

Line 388 – 389: is this the only result from CPF model? Please explain the reason to adopt this model?

Line 391-393: Needs more detail explanations to clarify.

Line 434: is it "TPSCF"?

Line 434 – 436: the author mentioned that CPF only can provide high probabilities from the west of the site. Please delete the CPF in this sentence.

Line 436 – 437: same sentence as Line 412 – 413. Please rephrase or rewrite.

---

## Author Comment (AC1) · 16 Jun 2016

The comment was uploaded in the form of a supplement:
http://www.atmos-chem-phys-discuss.net/acp-2015-939/acp-2015-939-AC1-supplement.zip

---

## Author Response (AR1)

June 18, 2016

Dear Editor,

We appreciate the reviewers' suggestions which have considerably improved the manuscript (**acp-2015-939**). Enclosed are point-by-point responses to the reviewers. We hope that with these changes the manuscript will be suitable for publication in "**Atmospheric Chemistry**

**and Physics**"

Thank you very much.

Sincerely,

Seung-Muk Yi

Professor, Dept. of Environmental Health, Graduate School of Public Health

Seoul National University, 1 Gwanak-ro, Gwanak-gu, Seoul 151-742, South Korea

Telephone: (82) 2-880-2736, Fax: (82) 2-762-9105, E-mail: yiseung@snu.ac.kr

**Response to Anonymous Referees' Comments**

● Journal: ACP

● Title: Characteristics of total gaseous mercury (TGM) concentrations in an industrial complex in southern Korea: Impacts from local sources

● Author(s): Yong-Seok Seo, Seung-Pyo Jeong, Thomas M. Holsen, Young-Ji Han, Eunhwa Choi, Eun Ha Park, Tae Young Kim, Hee-Sang Eum, Dae Gun Park, Eunhye Kim, Soontae Kim, Jeong-Hun Kim, Jaewon Choi, Seung-Muk Yi

● MS No.: acp-2015-939

● MS Type: Research article

● Status: Final Response

● Special Issue: Data collection, analysis and application of speciated atmospheric mercury

**Response to Anonymous Referee #1:**

**General comments**

This study analyzed seasonal and diurnal variations of TGM at a sampling site in southern Korea. Sources of TGM affecting the sampling site were investigated by correlating TGM with other pollutants and meteorological data and applying several source-receptor methods utilizing wind direction and back trajectory data. A newer method called the conditional bivariate probability function (CBPF) was used in this study to identify sources of TGM. However, I did not find this method very effective at differentiating between ground and stack emission sources. I find that there are many uncertainties in the CBPF results as well as in the back trajectory results that haven't been addressed in this paper. I have concerns about the methodology (insufficient TGM data, selection of trajectory duration) and interpretation of correlation analysis results. A discussion of how the results vary in the different seasons was also lacking in many places of the paper, even though the results are shown in the figures. Overall, I find that a major revision of this paper is necessary.

**Response**

Thank you for your comments. We believe we have addressed your concerns in the detailed responses below.

**Specific comments**

**Comment 1**

Line 57 – the correlation coefficient of r = -0.08 is very small. It's more accurate to state there is little correlation between TGM and air temperature

**Response 1**

As suggested, we have rephrased the sentence as follows on Line 56-57.

*"There was a **weak but** statistically significant negative correlation between the TGM concentration and ambient air temperature (r = -0.08) (p < 0.05)".*

**Comment 2**

Line 84 – "Atmospheric Hg released from natural and anthropogenic sources leading to enhanced deposition" Please clarify this statement.

**Response 2**

As suggested, we have rephrased the sentence as follows on Line 85-92.

**"***Atmospheric Hg released from natural and anthropogenic sources when introduced into*

*terrestrial and aquatic ecosystem through wet and dry deposition (Mason and Sheu, 2002) can*

*undergo various physical and chemical transformations before being deposited. Its lifetime in*

*the atmosphere depends on its reactivity and solubility so that depending on its form it can*

*have impacts on local, regional and global scales (Lin and Pehkonen, 1999; Lindberg et al.,*

*2007). A portion of the Hg deposited in terrestrial environments through direct industrial*

*discharge or atmospheric deposition is transported to aquatic system through groundwater and*

*surface water runoff (Miller et al., 2013).***"**

**Comment 3**

Lines 87-89 – Is this sentence about Hg emissions to the atmosphere and the biogeochemical cycling of Hg or the direct release of Hg from industrial effluent?

**Response 3**

We mean "the direct release of Hg from industrial effluent. In order to more clarify, we corrected "processes" to "**effluent**" on Line 93-95.

*"A previous study also reported that Hg directly released into terrestrial and aquatic*

*ecosystems from industrial **effluent** has influenced surface water, sediment and biological*

*tissue (Flanders et al., 2010)."*

**Comment 4**

Lines 94 – "coal combustion and waste incinerators" was already mentioned in this sentence.

**Response 4**

As suggested, we have rephrased the sentence as follows on Line 97-100.

"…*including municipal waste incinerators, medical waste incinerators, electric power*

*generating facilities and cement kilns (Dvonch et al., 1998), ferrous and non-ferrous metal*

*processing, iron and steel manufacturing facilities, and oil and coal combustion (Hoyer et al.,*

*1995)."*

**Comment 5**

Line 108 – use "data" instead of "information"

**Response 5**

As suggested, we corrected "information" to "data" as follows on Line 114.

"…combined with concentration **data** were used to identify…"

**Comment 6**

Line 138 – What temperature was the heated sampling line maintained at? Why is a heated sampling line necessary for sampling TGM?

**Response 6**

In this study, the sampling line was heated at about **50 ºC** using heat tape to prevent condensation in gold traps because moisture on gold surface interferes with the amalgamation of Hg.

In order to more clarify, we have rephrased the sentence as follows on Line 142-147.

*"Ambient air at a flow rate of 1.5 L $min^{-1}$ was transported through a 3 m-long heated sampling*

*line (1/4" OD Teflon) in to the analyzer. The sampling line was heated at about **50 ºC** using*

*heat tape to prevent water condensation in the gold traps because moisture on gold surfaces*

*interferes with the amalgamation of Hg (Keeler and Barres, 1999). Particulate matter was*

*removed from the sampling line by a 47 mm Teflon filter."*

**Comment 7**

2.3 QA/QC – the measurements were made for a one week period in each season. How often were the manual injections performed? Was there any maintenance activities performed prior to re-deployment of the instrument each time? These are important QA/QC procedures to mention because the instruments were offline for a long period of time.

**Response 7**

The reviewer is correct. In this study, manual injections were performed prior to every field sampling campaign and we continuously operated Tekran 2537B to measure GEM

concentrations in the ambient air.

In order to avoid any confusion, we have rephrased the sentence as follows on Line 159-160.

"*Manual injections were performed prior to every field sampling campaign to evaluate these*

*automated calibrations using a saturated mercury vapor standard.*"

**Comment 8**

Line 168 – This definition of CPF doesn't seem right because it is not exactly the source contribution. You can replace this sentence with the one in line 177.

**Response 8**

The reviewer is correct. We have rephrased the sentence as follows on Line 171-173.

"*CPF estimates the probability that the measured concentration will exceed the threshold*

*criterion for a given wind direction.*"

**Comment 9**

Lines 173-175 – were the wind data measured every 5 min similar to TGM or was it averaged to the nearest hour?

**Response 9**

In this study, **hourly** meteorological data (air temperature, relative humidity, and wind speed and direction) were obtained from the Automatic Weather Station (AWS) operated by the Korea

Meteorological Administration (KMA) (http://www.kma.go.kr).

In order to clarify, we added "hourly" in the sentence as follows on Line 150-152.

"***Hourly*** *meteorological data (air temperature, relative humidity, and wind speed and direction)*

*were obtained from the Automatic Weather Station (AWS) operated by the Korea*

*Meteorological Administration (KMA) (http://www.kma.go.kr) (6 km from the site).*"

This information was also described in Table 2 as follows on Line 539-540.

"***Table 2***. *Summary of atmospheric concentrations of TGM and co-pollutants, and*
*meteorological data. Note that TGM was measured every 5-min, and other pollutants and*
*meteorological data were measured every 1-hour.*"

**Comment 10**

Lines 183-185 –How does this method account for the full distribution of concentrations rather than concentrations exceeding a threshold? Based on Eq. 2, CBPF analyzes the subset of concentrations above a threshold as well. Another thing is how does this method account for sources with different dispersion characteristics? The equation is based on horizontal wind speeds, which is advection rather than dispersion.

**Response 10**

The statement in question is incorrect since we used a criteria to determine which concentrations to include so we have deleted the sentence.

**Comment 11**

Lines 193-195 – This explanation is not clear. Can you give some examples of mercury sources with different wind speed dependencies?

**Response 11**

In order to more clarify, we rephrased the sentence as follows on Line 195-198.

"*The extension to the bivariate case can provide more information on the nature of the sources because different source types such as stack emission sources and ground-level sources can have different wind speed dependencies (prominent at low and high wind speed).*"

**Comment 12**

Lines 201-215 – this part needs to be rewritten by improving on the wording

**Response 12**

As suggested, we rephrased the sentence as follows on Line 204-218.

"*The PSCF is a simple method that links residence time in upwind areas with high concentrations through a conditional probability field and was originally developed by*

*Ashbaugh et al. (1985).*

*PSCF$_{ij}$ is the conditional probability that an air parcel that passed through the ijth cell had a*

*high concentration upon arrival at the monitoring site and is defined as the following Eq. (3).*

$$PSCF_{ij} = \frac{m_{ij}}{n_{ij}} \qquad (3)$$

*where, $n_{ij}$ is the number of trajectory segment endpoints that fall into the ij-th cell, and $m_{ij}$ is the*

*number of segment endpoints in the same grid cell (ij-th cell) when the concentrations are higher*

*than a criterion value as measured at the sampling site.*

*High PSCF values in those grid cells are regarded as possible source locations. Cells including*

*emission sources can be identified with conditional probabilities close to one if trajectories that*

*have crossed the cells efficiently transport the released pollutant to the receptor site. Therefore, the*

*PSCF model provides a tool to map the source potentials of geographical areas."*

**Comment 13**

Line 220 –Please justify the use of 24 hr trajectories. TGM is mainly GEM which has a longer residence time and capable of long range transport. This means a longer trajectory duration would be more suitable.

**Response 13**

In this study, 24h backward trajectories starting at every hour at a height of 10, 50 and 100 m above ground level were computed using the vertical velocity model because we identified the diurnal variations in TGM concentrations are due to a combination of 1) reactions with an oxidizing atmosphere, 2) changes in ambient temperature and 3) local emissions related to industrial activities. This information was described in Line 428-430.

Previous studies reported that identified the long-range transport of mercury using the

ΔTGM/ΔCO enhancement ratio (Choi et al., 2009; Jaffe et al., 2005; Kim et al., 2009; Weiss-

Penzias et al., 2003; Weiss-Penzias et al., 2006). The observed ΔTGM/ΔCO was significantly lower than that of Asian long-range transport, but similar to that of local sources in Korea and in US industrial events suggesting that local sources are more important than that of long-range transport in this study. This information was also described in Line 336-352.

Based on the above results, PSCF was performed to identify the local sources over grid cells corresponding to Gyeongsangbuk-do in eastern South Korea.

In addition, we did not find significant differences between TPSCF using 24 h and 48 h backward trajectories (Fig R1).

[Figure]

Fig. R1. Comparisons of TPSCF results using 24 h (top) and 48 h (bottom) backward trajectory.

In order to clarify, we have rephrased the sentence as follows on Line 223-227.

*"As will be discussed in Section 5.4, 24 h backward trajectories starting at every hour at a*

*height of 10, 50, and 100 m above ground level were computed using the vertical velocity model*

*because local sources are more important than that of long-range transport in this study (It*

*should be noted that PSCF results using 48 h backward trajectories had similar results as the*

*24 h backward trajectories)."*

**Comment 14**

Line 229 – should be "meteorological data"

**Response 14**

As suggested, we corrected "meteorology data" to "**meteorological data**" as follows on Line

235.

*"PSCF was calculated with 9 km **meteorological data**"*

**Comment 15**

Lines 293-294 – This point was not discussed in later sections

**Response 15**

We have deleted the phrase "*As will be discussed later*" and rephrased as follows on Line 312-

314.

*"The high TGM concentrations in fall was due to different wind direction (see Fig. S2), sources,*

*relationships with other pollutants and meteorological conditions. More detailed information*

*can be found in Section 5.5."*

**Comment 16**

Lines 299-302 – It should be stated more clearly that combustion was not a source of TGM

because of a lack of correlation between TGM and the other combustion pollutant markers.

**Response 16**

As suggested, we have rephrased the sentence as follows on Line 319-321.

"*However, the TGM concentration was not significantly correlated with NO2, CO or SO2*

*concentrations, suggesting that combustion associated with space heating was not a significant*

*source of TGM (Choi et al., 2009)*"

**Comment 17**

Lines 315 – Fig. S4 shows the CPF and CBPF plots in each season. Should this be discussed in section 5.5? It's not clear how these plots relate to the correlation results.

    **Response 17**

In Section 5.3, we investigated the correlation between TGM and CO and found that local sources are more important than that of long-range transport by the observed ΔTGM/ΔCO. In order to avoid confusion, we have deleted Fig. S4 and rephrased the sentence as follows on

Line 332 to Line 335.

*"On the other hand, there were no statistically significant correlations between TGM and CO*

*in spring (r = 0.02) (p = 0.78), in summer (r = 0.13) (p = 0.08), or in fall (r = -0.03) (p = 0.69),*

*indicating that TGM and CO were affected by different anthropogenic emission sources in these*

*seasons."*

**Comment 18**

Lines 326-333 – It's surprising that long-range transport from China did not impact this site considering that it affected elevated Hg events in Seoul, Japan, and North America in previous studies. I would think long-range transport impacts a larger region including this sampling location. The TGM/CO slopes during elevated Hg events in Seoul were attributed to both long- range transport and local source impacts (Choi et al., 2009). Is it possible that the one-week sampling period in each season did not capture the long-range transport events? More data is needed to confirm these results. There are also uncertainties from the potential mixing between long-range transported airflows and local air making it difficult to distinguish between distant and local source impacts.

    **Response 18**

We agree with the reviewer's comment. We also think it is possible that the one-week sampling period in each season did not capture the long-range transport events, and more data are needed to confirm these results. Based on the previous studies (Kim et al., 2009; Choi et al., 2009), we analyzed the TGM data in an attempt to identify both long-range transport and local sources of

TGM. Unfortunately, we did not find high concentration events which were defined as at least a 10 h period with hourly average TGM and CO concentrations higher than the average monthly TGM and CO concentrations and high values of TGM/CO ratio ($\Delta$TGM/$\Delta$CO)

(0.0052-0.0158 ng m$^{-3}$ ppb$^{-1}$) and high correlations (r$^2$>0.5) in this study. Therefore, the observed $\Delta$TGM/$\Delta$CO suggested that local sources are more important than that of long-range transport in this study. However, we believe more can be learned using the larger dataset than just using the one-week sampling period.

In order to any confusion, we added a following sentence on Line 353 to Line 357.

*"There are also uncertainties from the potential mixing between Hg associated with long-range*

*transported airflows and local air making it difficult to distinguish between distant and local*

*source impacts. However, it is possible that the one-week sampling period in each season did*

*not capture the long-range transport events, and more can be learned using a larger dataset*

*than just using the one-week sampling period to confirm these results."*

**Comment 19**

Lines 345-346 – what are the time periods for daytime and nighttime concentrations? This sentence states daytime TGM were higher during daytime than nighttime. But in the previous sentence, the minimum TGM concentration occurs in the afternoon.

**Response 19**

As suggested, we added the time periods for daytime and nighttime as follows on Line 361-

363.

*"TGM, $O_3$, CO, $SO_2$, and temperature in the daytime (__06:00-18:00__) were statistically*

*significantly higher than those in the nighttime (__18:00-06:00__) (p < 0.05) except $PM_{10}$ (p =*

*0.09)..."*

In this study, the daytime TGM concentration (5.3 ± 4.7 ng m$^{-3}$) was statistically significantly higher than that in nighttime (4.7 ± 4.7 ng m$^{-3}$) (p < 0.01). In order to avoid any confusion, we corrected "maximum" to "**increase**" and "minimum" to "**decrease**" as follows on Line 365 to Line 366.

*"TGM generally showed a consistent diurnal variation with an **increase** in the early morning (06:00-09:00) and a **decrease** in the afternoon (14:00-17:00), similar to…"*

**Comment 20**

Lines 353-355 – You discussed about the land-sea/lake breeze effect on TGM diel patterns from another study in the previous sentence. Does this atmospheric process affect this particular site since it is near the ocean and lower TGM were also observed during daytime?

**Response 20**

It is possible that the land-sea breeze might affect diurnal variations in TGM concentrations since the sampling site was near the ocean and lower TGM concentrations were observed during the daytime in this study. However, there are several known Hg sources such as iron and steel manufacturing facilities including electric and sintering furnaces using coking around the sampling site.

Although the daytime temperature ($14.7 \pm 10.0$ ℃) was statistically significantly higher than that in the nighttime ($13.0 \pm 9.8$ ℃) ($p < 0.05$), the daytime TGM concentration ($5.3 \pm 4.7$ ng m$^{-3}$) was statistically significantly higher than those in the nighttime ($4.7 \pm 4.7$ ng m$^{-3}$) ($p < 0.01$). This is possibly due to a combination of 1) reactions with an oxidizing atmosphere, 2) changes in ambient temperature and 3) local emissions related to industrial activities. To supplement these conclusions CPF and CBPF were used to identify source directions and TPSCF was used to identify potential source locations in "Section *5.6 CPF, CBPF and TPSCF results of TGM*".

In order to clarify, we rephrased the sentence as follows on Line 382 to Line 390.

*"Although it is possible that the land-sea breeze may affect diurnal variations in TGM concentrations since the sampling site was near the ocean and lower TGM were also observed during the daytime, the higher concentrations in the daytime than those in nighttime were due to local emission sources because the daytime temperature (14.7 ± 10.0 ºC) was statistically significantly higher than that in the nighttime (13.0 ± 9.8 ºC) (t-test, $p < 0.05$) and there was a weak but statistically significant negative correlation between TGM concentration and ambient air temperature (r = -0.08) ($p < 0.05$). In addition, there are several known Hg sources such as iron and steel manufacturing facilities including electric and sintering furnaces using coking between the sampling site and the ocean."*

**Comment 21**

Lines 355-356 – The negative correlation between TGM and temperature is very small (as mentioned in the abstract, r = -0.08) despite a significant p-value. It's more accurate to state there is little relationship between TGM and air temperature.

  **Response 21**

As suggested, we have rephrased the sentence as follows on Line 386-388.

*"… and there was a weak but statistically significant negative correlation between the TGM concentration and ambient air temperature (r = -0.08) ($p < 0.05$)".*

**Comment 22**

Lines 357-360 – Similar to the above comment, the correlation between TGM and O3 is too small (r = -0.18) to suggest that it is indicative of GEM oxidation. It's more correct to state there is little relationship between TGM and O3. If GEM oxidation occurred, GOM concentrations would increase. There are some uncertainties on the net effect on TGM (GEM+GOM).

  **Response 22**

As suggested, we have rephrased the sentence as follows on Line 391-399.

*"...there was a weak but negative relationship between the TGM concentrations and O3*

*concentrations (r = -0.18) (p < 0.01), suggesting that oxidation of GEM in the oxidizing*

*atmosphere during periods of strong atmospheric mixing was partially responsible for the*

*diurnal variations of TGM concentrations. In addition, oxidation of GEM by bromine species*

*in the coastal area (Obrist et al., 2011) or by chloride radicals in marine boundary layer*

*(Laurier et al., 2003) might play a significant role. If oxidation of GEM occurred, GOM*

*concentrations would increase. However there are uncertainties on the net effects on TGM (the*

*sum of the GEM and the GOM) since we did not measure GOM concentrations."*

**Comment 23**

Lines 369-378 – The results here are inconsistent. If the small negative correlation between

TGM and temperature indicates an increase in mixing height which leads to a decrease in TGM, how can it explain surface emissions in the morning which should increase with temperature?

Is there a positive correlation between TGM and temperature in the morning?

**Response 23**

As suggested, we analyzed a relationship between TGM and temperature in the morning (06:00-09:00) and found that there is a positive correlation between TGM and ambient air temperature. In order to clarify, we added a following sentence on Line 414-418.

*"Nonparametric correlations revealed that there is a positive correlation between TGM and*

*ambient air temperature ($r_s = 0.11$, p=0.27) between 06:00-09:00. The TGM concentration*

*was negatively correlated with $O_3$ ($r_s = -0.33$, p<0.01) but positively correlated with $NO_2$ ($r_s*

*= 0.21$, p<0.05), suggesting that the increased traffic is the main source of TGM during these*

*time periods."*

**Comment 24**

Section 5.4 – This section is lacking discussion on seasonal differences in the diurnal variation.

The fall diurnal pattern appears very different from those in other seasons in Fig. 3. Can you discuss why the TGM were much higher overnight in the fall but daytime concentrations were similar to those in other seasons? Why was there a large drop in TGM from 4:00 to 5:00 in the fall?

**Response 24**

As suggested, we described more detailed information on Line 419-427 as follows.

*"Compared to other seasons, significantly different diurnal variations of TGM were observed*

*in fall. The daytime TGM concentrations in fall were similar to those in other seasons, however,*

*the nighttime TGM concentrations in fall were much higher than other seasons. As described*

*earlier in Section 5.3, the high TGM concentrations in fall was possibly due to the relationship*

*between other pollutants and meteorological conditions as well as different wind direction and*

*sources. The nighttime TGM concentrations in fall were simultaneously positively correlated*

*with $PM_{10}$ (r=0.26) (p<0.05) and CO (r=0.21) (p<0.05) concentrations and wind speed*

*(r=0.35) (p<0.01), suggesting that the combustion process is an important source during this*

*period."*

We could not identify the reason why there was a large drop in TGM from 4:00 to 5:00 in the fall because there were a limited amount of data. However, we believe more can be learned using a larger dataset.

**Comment 25**

Lines 381-383 – These conclusions are not well-supported by the correlation analyses because the correlation coefficients were very small.

**Response 25**

As suggested, we rephrased the "Section 5.5 *Diurnal variations*" on Line 359-432.

**Comment 26**

Line 390 – What are the potential Hg sources from the northeast direction?

**Response 26**

The potential Hg sources from the northeast direction is due to lots of domestic passenger ships routes. In response to this comment, we have rephrased the sentence as follows on Line 475-

477.

*"The northeast direction including the East Sea was also identified as potential source areas*

*likely because this is an area with lots of domestic passenger ships routes."*

**Comment 27**

Lines 391-393 – Please clarify this sentence. Are the higher wind speeds associated with stack emissions and lower wind speeds associated with surface emissions? There are several issues or uncertainties with this point. (1) Wind speeds were only measured at one height. How can you tell that the lower wind speeds are from lower elevation and vice-versa? What is the height of the wind measurements? (2) As mentioned in the diurnal variation section, boundary layer mixing occurs during the day. Is it possible to distinguish between ground emissions and stack emissions? (3) It seems only the west directions had both high and low wind speeds, while the east directions had only lower wind speeds (Fig. S1). Thus, CBPF doesn't seem useful when there is a lack of wind speed variation. It appears that it is by coincidence that both ground level and stack emissions were identified in the west direction because there happened to be a wind speed variation from this direction. Based on these points, the CBPF results don't seem to reveal more about TGM sources than CPF. More discussion is needed on the relationship between specific sources and wind speeds. Instead of wind speed, what other variables would be useful for source identification using CBPF?

**Response 27**

In this study, hourly meteorological data (air temperature, relative humidity, and wind speed and direction) were obtained from the Automatic Weather Station (AWS) operated by the Korea Meteorological Administration (KMA) (http://www.kma.go.kr). This information was described on Line 150-152. We used the wind data measured at a reference height of 10 m.

Although there is a lack of wind speed variation, we found that the CBPF revealed more about TGM sources than CPF (Fig. S6).

The third variable plotted on the radial axis does not need to be wind speed. A previous study reported that temperature can be a useful radial variable (Carslaw, D.C., Beevers, S.D., 2013. Characterising and understanding emission sources using bivariate polar plots and k-means clustering. Environ. Model. Softw. 40 (0), 325-329).

In order to response to this comment, we rephrased the sentence on Line 195-198 and Line 433-450 as follows.

< Line 195-198>

*"The extension to the bivariate case can provide more information on the nature of the sources*

*because different source types such as stack emission sources and ground-level sources can*

*have different wind speed dependencies (prominent at low and high wind speed)."*

< Line 439-456>

*"CBPF shows that the high probabilities from the west occurred under high wind speed (>*

*3 m s⁻¹) indicative of emissions from stacks as well as low wind speed (≤3 m s⁻¹) indicative of*

*non-buoyant ground level sources.*

*As described in Section 5.4, correlations between TGM and CO revealed that TGM and CO*

*were affected by similar anthropogenic emission sources in winter but affected by different*

*sources in spring, summer and fall, which is supported by Fig. S6 which shows significantly*

*different seasonal patterns of CPF and CBPF for TGM concentrations.*

*It is difficult to discuss about the different seasonal patterns for CPF and CBPF for TGM*

*concentrations since there were no correlations between TGM and other pollutants in spring,*

*summer and fall except $O_3$. However, compared to Fig. 4, the CPF and CBPF patterns in fall*

*were similar to those during the whole sampling periods. Especially, the nighttime TGM*

*concentration in fall was simultaneously positively correlated with $PM_{10}$ (r=0.26) (p<0.05)*

*and CO (r=0.21) (p<0.05) concentrations and wind speed (r=0.35) (p<0.01), indicating that*

*the combustion process from the west is an important source during this period.*

*Since TGM showed a significant correlation with CO (r=0.25) (p<0.05) and showed a weak*

*positive correlation with $PM_{10}$ (r=0.08) (p=0.33) in winter with high wind speed, combustion*

*sources from the west are likely partially responsible for this result."*

**Comment 28**

Lines 394-395 – Are there industrial sources south of the sampling site? High probability areas are also identified in this direction in the TPSCF plot in Fig. 4.

**Response 28**

There is Ulsan Metropolitan City located to the south from the sampling site. Ulsan is South

Korea's seventh largest metropolis with a population of over 1.1 million and it has more than
700 small and large industrial facilities including petrochemical plants, oil refineries, vehicle
and ship factories, and other chemical plants.
In response to this comment, we have added a following sentence on Line 477 to Line 480.
*"The south from the sampling was also identified as a likely source area of TGM where Ulsan*
*Metropolitan City, South Korea's seventh largest metropolis with a population of over 1.1*
*million is located. It includes a large petrochemical complex known as a TGM source (Jen et*
*al., 2013)."*
**Comment 29**
Fig. 4 – the source areas seem confined to the industrial complex near the sampling site because
of the short trajectory duration (24 hrs). Use of longer trajectories would help expand the source
region and identify potential regional transport to the site. In addition to this uncertainty, other
PSCF uncertainties should be discussed.
  **Response 29**
As mentioned earlier in **Response 13**, local sources are more important than that of long-range
transport in this study. Therefore PSCF was performed to identify the local sources over grid
cells corresponding to Gyeongsangbuk-do in eastern South Korea.
In addition, we did not find significant differences between PSCF using 24 h and 48 h backward
trajectories (Fig. R1 in **Response 13**).
In response to this comment, we added a following sentence on Line 240-245.
*"Previous studies suggest that there are increasing uncertainties as backward trajectory*
*distances increase (Stohl, et al., 2002) and that PSCF modeling is prone to the trailing effect*
*is which locations upwind of sources are also identified as potential sources (Han, et al., 2004).*
*An alternative to back trajectory calculations in the interpretation of atmospheric trace*
*substance measurements (Stohl, et al., 2002) although this technique does not provide much*
*information on source locations"*
**Comment 30**

Section 5.5 – This section is lacking discussion on the CPF and CBPF results in different
seasons. The seasonal plots are shown in Fig. S4, but they are not discussed in this section.
**Response 30**
In order to response to this comment, we added the sentence as follows on Line 443-456.
*"As described in Section 5.4, correlations between TGM and CO revealed that TGM and CO*
*were affected by similar anthropogenic emission sources in winter but affected by different*
*sources in spring, summer and fall, which is supported by Fig. S6 which shows significantly*
*different seasonal patterns for CPF and CBPF for TGM concentrations. It is difficult to*
*interpret differences in the seasonal patterns of CPF and CBPF results for TGM concentrations*
*since there were no correlations between TGM and other pollutants in spring, summer and fall*
*except $O_3$. However, compared to Fig. 4, the CPF and CBPF patterns in fall were similar to*
*those during the complete sampling period. The nighttime TGM concentration in fall was*
*simultaneously positively correlated with $PM_{10}$ (r=0.26) (p<0.05) and CO (r=0.21) (p<0.05)*
*concentrations and wind speed (r=0.35) (p<0.01), indicating that the combustion process from*
*the west is an important source during this period. Since TGM showed a significant correlation*
*with CO (r=0.25) (p<0.05) and showed a weak positive correlation with $PM_{10}$ (r=0.08)*
*(p=0.33) in winter with high wind speed, combustion sources from the west are likely partially*
*responsible for this result."*

[Figure]

**Fig. S6. Comparisons of CPF and CBPF plots for TGM and CO higher than average concentration. The radial axes of CPF and CBPF are the probability and the wind speed (m s$^{-1}$), respectively.**

**Comment 31**

Lines 421-423 – This conclusion was not discussed in the results section. The wind direction frequency plots in different seasons are shown in Fig. S1 but the results were not discussed in the paper.

**Response 31**

As suggested, we rephrased the sentence on Line 419-427 as follows.

 **Comment 32**

 Lines 427-430 – p-value was significant but the correlation coefficient (magnitude of the

 relationship) is too small. I also don't understand the logic in these results. There should be a

 positive TGM and temperature correlation if daytime TGM concentrations were related to

 surface emissions.

          **Response 32**

 In response to this comment, we have rephrased the sentence on Line 369-377 and Line 382- as follows.

 < Line 369-377>

 *"The daytime TGM concentration (5.3 ± 4.7 ng m$^{-3}$) was higher than that in nighttime (4.7 ±*

 *4.7 ng m$^{-3}$) (p < 0.01), which was similar to several previous studies (Cheng et al., 2014;*

 *Gabriel et al., 2005; Nakagawa, 1995; Stamenkovic et al., 2007) but different than other study*

 *(Lee et al., 1998). Previous studies reported that this is due to local sources which is close to*

 *the sampling site (Cheng et al., 2014; Gabriel et al., 2005), positive correlation between TGM*

 *concentration and ambient air temperature (Nakagawa, 1995) and increased traffic*

 *(Stamenkovic et al., 2007). However, other study suggested that the higher TGM concentration*

 *than that in daytime was due to the shallow boundary layer in nighttime, resulting in*

 *concentrating the TGM near the surface (Lee et al., 1998)."*

 < Line 382-390>

 **"***Although it is possible that the land-sea breeze may affect diurnal variations in TGM*

 *concentrations since the sampling site was near the ocean and lower TGM were also observed*

 *during the daytime, the higher concentrations in the daytime than those in nighttime were due*

 *to local emission sources because the daytime temperature (14.7 ± 10.0 ℃) was statistically*

 *significantly higher than that in the nighttime (13.0 ± 9.8 ℃) (t-test, p < 0.05) and there was a*

 *weak but statistically significant negative correlation between TGM concentration and ambient*

 *air temperature (r = -0.08) (p < 0.05). In addition, there are several known Hg sources such*

 *as iron and steel manufacturing facilities including electric and sintering furnaces using coking*

 *around the sampling site.*"

 **Comment 33**

 Line 434 – JP-PSCF was not used anywhere else in the paper. Did you mean TPSCF?

**Response 33**

The reviewer is correct. As suggested, we corrected "JP-PSCF" to "TPSCF" as follows on Line

503.

*"However, CBPF and TPSCF indicated that…"*

**Response to Anonymous Referee #2:**

**General comments**

This study measured TGM concentrations in South Korea and analyzed seasonal and diurnal variations of TGM. They also used the statistics analysis to correlate TGM with other pollutants and meteorological data. They tried to identify the possible TGM sources using CPF, CBPF, and TPSCF models. They found that the nearby local sources are more significant than others. Over all, this paper meets the original contributions and contains unique TGM data nearby industrial areas. The authors performed the appropriate modeling approaches to identify possible mercury sources. However, the presentation quality doesn't meet the ACP's requirement. I suggest that the paper should be carefully revised and edited prior to publication on ACP.

Comment

   **Response**

Thank you for your comments. We carefully revised and edited that paper as suggested as is detailed in our responses to the specific comments.

**Specific comments**

Introduction section contains too basic and out of dated information. I suggest adding some recent mercury inventory/modeling studies in East Asia. Result and discussion contains unnecessarily much literature review. More discussions are needed. The author used statistics analysis in many places. Please provide the type of analysis in this paper.

**Comment 1**

Line 118 – 133: it should be combined into one paragraph.

   **Response 1**

As suggested, the sentence was combined into one paragraph as follows on Line 125-138.

**Comment 2**

Line 133: So what are their results in Kim et al. 2010? What did they find?

   **Response 2**

This meant that we got and used the Hg emissions data from the authors (Kim et al., 2010)

about iron and steel manufacturing, and a hazardous waste incinerator.

In order to clarify, we have revised the sentence as follows on Line 137 to Line 138.

*"Hg emissions __data__ from iron and steel manufacturing, and a hazardous waste incinerator*

*were estimated __based on__ a previous study (Kim et al., 2010)"*

Their results in Kim et al. (2010) were described in Line 104-109.

**Comment 3**

Line 146-151: it doesn't fit in material & method section. Please move to results & discussion section.

    **Response 3**

As suggested, we moved that sentence to "5. Results and Discussions" on Line 281-286 as follows.

*"5.1. Meteorological data analysis*

*Fig. S2 shows the frequency of counts of measured wind direction occurrence by season*

*during the sampling period. The predominant wind direction at the sampling site was W (20.9%)*

*and WS (19.2%), and calm conditions of wind speed less than 1 m s$^{-1}$ occurred 7.6% of the*

*time. Compared to other seasons, however, the prevailing winds in summer were N (17.0%),*

*NE (16.4%), S (16.4%), and SW (15.8%)."*

**Comment 4**

Line 156-157: need to explain how often manual injections were conducted.

    **Response 4**

Manual injections were performed prior to every field sampling campaign and we continuously operated Tekran 2537B to measure GEM concentrations in the ambient air.

In order to avoid any confusion, we have rephrased the sentence as follows on Line 159 to Line

160.

*"Manual injections were performed prior to every field sampling campaign to evaluate these*

*automated calibrations using a saturated mercury vapor standard."*

**Comment 5**

Line 164: already used CPF in Line 163. Replace "Conditional Probability Function (CPF)" with "CPF"

**Response 5**

As suggested, "Conditional Probability Function (CPF)" was replaced with CPF on Line 168.

**Comment 6**

Line 220 – 222 and line 230 – 233 are same.

**Response 6**

The sentence on Line 220-222 means that we computed 24hr backward trajectories starting at every hour at a height of 10, 50, and 100 m above ground level.

The sentence on Line 230-233 means that TPSCF which incorporates probability from **above**

**different starting heights (10, 50, and 100 m above ground level)** was calculated.

In order to more clarify, we rephrased the sentence as follows on Line 236-237.

*"In this study, TPSCF which incorporates probability from above different starting heights was*

*calculated…"*

**Comment 7**

Line 237: what is $n_{ij}$ values here?

**Response 7**

$n_{ij}$ is the number of trajectory segment endpoints that fall into the *ij*-th cell. This information was described in the sentence on Line 212.

In order to clarify, we added "the number of trajectory segment endpoints that fall into the *ij-*

th cell" to the sentence on Line 249 as follows.

*"To minimize the effect of small $n_{ij}$ (the number of trajectory segment endpoints that fall into*

*the ij-th cell) values, resulting in high TPSCF values ...."*

**Comment 8**

Line 272-275: Can the author provide recent TGM data from China and other country?

> **Response 8**

As suggested, we added recent TGM data from China and other countries in Table 1 on Line and rephrased the sentence on Line 293-294.

*"...and those measured in China, in Japan and other locations in Korea, however considerably*

*lower than those measured near large Hg sources in Guangzhou, China (Table 1)."*

**Comment 9**

Line 285 and later: if the author mentioned p-value ($p < 0.01$ or $p < 0.05$), "statistically significantly" does not have to be addressed every time. Readers already know that the author performed statistical analysis.

> **Response 9**

After Line 285, we deleted the phrase "statistically significantly" as suggested (after Line 304).

**Comment 10**

Line 293: "as will be discussed later: : :." Can you indicate where and which section it was discussed?

> **Response 10**

We have deleted the phrase "*As will be discussed later*" as follows on Line 312.

*"The high TGM concentrations in fall was due to ..."*

**Comment 11**

This 5.4 section is for result and discussion. It includes too much literature review rather than discussion.

**Response 11**

As suggested, we rephrased the 5.4 section as following Section "*5.5. Diurnal variations*" on

Line 359-432.

**Comment 12**

Line 346 – 348: can you explain what previous studies concluded about these diurnal variations?

Needs more discussion.

**Response 12**

As suggested, we rephrased the sentence on Line 369-377 as follows.

*"The daytime TGM concentration ($5.3 \pm 4.7$ ng m$^{-3}$) was higher than that in the nighttime ($4.7$*

*$\pm 4.7$ ng m$^{-3}$) ($p < 0.01$), which was similar to several previous studies (Cheng et al., 2014;*

*Gabriel et al., 2005; Nakagawa, 1995; Stamenkovic et al., 2007) but different than another*

*studies (Lee et al., 1998). Previous studies reported that this different is due to local sources*

*close to the sampling site (Cheng et al., 2014; Gabriel et al., 2005), a positive correlation*

*between TGM concentration and ambient air temperature (Nakagawa, 1995) and increased*

*traffic (Stamenkovic et al., 2007). However, another study suggested that the higher TGM*

*concentration during the night was due to the shallowing of the boundary layer, which*

*concentrated the TGM near the surface (Lee et al., 1998)"*

**Comment 13**

Line 353: "as will be discussed later…" Indicate where and which section it was discussed?

What is "this" mean here? Does "this" mean lower TGM in daytime?

**Response 13**

In order to clarify, we have rephrased the sentence on Line 382-390 as follows.

**"***Although it is possible that the land-sea breeze may affect diurnal variations in TGM*

*concentrations since the sampling site was near the ocean and lower TGM were also observed*

*during the daytime, the higher concentrations in the daytime than those in nighttime were due*

*to local emission sources because the daytime temperature ($14.7 \pm 10.0$ ℃) was statistically*

*significantly higher than that in the nighttime ($13.0 \pm 9.8$ ℃) (t-test, $p < 0.05$) and there was a*

*weak but statistically significant negative correlation between TGM concentration and ambient*

*air temperature (r = -0.08) (p < 0.05). In addition, there are several known Hg sources such*

*as iron and steel manufacturing facilities including electric and sintering furnaces using coking*

*between the sampling site and the ocean.*"

**Comment 14**

Line 355 – 356 and line 369 – 370 are same. Please rephrase or rewrite.

**Response 14**

As suggest, we rephrased the sentence as follows on Line 406-407.

*"TGM concentration was negatively correlated with ambient air temperature (r = -0.08) (p <*

*0.05) because high ambient air temperature…"*

**Comment 15**

Line 369 – 378: this paragraph is vague. Please clarify.

**Response 15**

In response to this comment, we added a following sentence as follows on Line 414-418.

*"Nonparametric correlations revealed that there is a positive correlation between TGM and*

*ambient air temperature ($r_s$ = 0.11, p=0.27) between 06:00-09:00. The TGM concentration*

*was negatively correlated with $O_3$ ($r_s$ = -0.33, p<0.01) but positively correlated with $NO_2$ ($r_s$*

*= 0.21, p<0.05), suggesting that the increased traffic is the main source of TGM during these*

*time periods."*

**Comment 16**

This 5.5 section also has too much literature review rather than discussion.

**Response 16**

As suggested, we rephrased the Section 5.5 as follows on Line 434-480 (see the Section *5.6.*

*CPF, CBPF and TPSCF results of TGM)*

**Comment 17**

Line 381 – 385: this paragraph is the result from the Section 5.4. Please move it to Section 5.4.

**Response 17**

As suggested, we moved the paragraph to Line 428 to Line 432.

**Comment 18**

Line 388 – 389: is this the only result from CPF model? Please explain the reason to adopt this model?

**Response 18**

We showed the CBPF result as well as CPF result. This information was described in the sentence on Line 439 to Line 442 as follows.

*"…CBPF shows that the high probabilities from the west occurred under high wind speed (>*

*3 m s⁻¹) indicative of emissions from stacks as well as low wind speed (≤ 3 m s⁻¹) indicative of*

*non-buoyant ground level sources (Uria-Tellaetxe and Carslaw, 2014)."*

**Comment 19**

Line 391-393: Needs more detail explanations to clarify.

**Response 19**

In order to clarify, we rephrased the sentence on Line 439 to Line 442 as follows.

*"CBPF shows that the high probabilities from the west occurred under high wind speed (> 3*

***m s⁻¹****) indicative of emissions from stacks as well as low wind speed (≤ **3 m s⁻¹***) indicative of*

*non-buoyant ground level sources (Uria-Tellaetxe and Carslaw, 2014)."*

**Comment 20**

Line 434: is it "TPSCF"?

**Response 20**

The reviewer is correct. As suggested, we corrected "JP-PSCF" to "TPSCF".

**Comment 21**

Line 434 – 436: the author mentioned that CPF only can provide high probabilities from the west of the site. Please delete the CPF in this sentence.

**Response 21**

In order to response to this comment, we rephrased the sentence as follows on Line 501-506.

*"CPF only shows high probabilities to the west from the sampling site where there are large*

*steel manufacturing facilities and waste incinerators. However, CBPF and TPSCF indicated*

*that the dominant sources of TGM were the hazardous waste incinerators and the coastal areas*

*in the northeast as well as the iron and manufacturing facilities in the west."*

**Comment 22**

Line 436 – 437: same sentence as Line 412 – 413. Please rephrase or rewrite.

**Response 22**

As suggested, we rephrased the sentence on Line 436-437 as a following sentence on Line 505

to 506.

[revised manuscript text omitted]

---

## Author Response (AR2)

July 17, 2016

Dear Editor,

We appreciate the reviewers' suggestions which have considerably improved the manuscript (**acp-2015-939**). Enclosed are point-by-point responses to the reviewers. We hope that with these changes the manuscript will be suitable for publication in "**Atmospheric Chemistry**

**and Physics**"

Thank you very much.

Sincerely,

Seung-Muk Yi

Professor, Dept. of Environmental Health, Graduate School of Public Health

Seoul National University, 1 Gwanak-ro, Gwanak-gu, Seoul 151-742, South Korea

Telephone: (82) 2-880-2736, Fax: (82) 2-762-9105, E-mail: yiseung@snu.ac.kr

**Response to Anonymous Referees' Comments**

● Journal: ACP

● Title: Characteristics of total gaseous mercury (TGM) concentrations in an industrial complex in southern Korea: Impacts from local sources

● Author(s): Yong-Seok Seo, Seung-Pyo Jeong, Thomas M. Holsen, Young-Ji Han, Eunhwa Choi,

Eun Ha Park, Tae Young Kim, Hee-Sang Eum, Dae Gun Park, Eunhye Kim, Soontae Kim, Jeong-

Hun Kim, Jaewon Choi, Seung-Muk Yi

● MS No.: acp-2015-939

● MS Type: Research article

● Status: File Upload (ACP)

● Iteration: Minor Revision

● Special Issue: Data collection, analysis and application of speciated atmospheric mercury

**Response to Anonymous Referee #1:**

**Comment 1**

Line 195: "The extension to the bivariate case can provide more information on the nature of the sources because different source types such as stack emission sources and ground-level sources can have different wind speed dependencies (prominent at low and high wind speed)." This statement is incorrect because wind speeds are higher at higher elevation than at ground level. It also contradicts a related sentence in lines 439-456: "CBPF shows that the high probabilities from the west occurred under high wind speed ($>3$ m s $-1$) indicative of emissions from stacks as well as low wind speed ($\leq 3$ m s 1) indicative of non-buoyant ground level sources"

**Response 1**

The reviewer is correct – we reversed the order in this sentence. The corrected version is shown below (Please see the Line 225-228).

*"The extension to the bivariate case can provide more information on the nature of the sources because different source types such as stack emission sources and ground-level sources can have different wind speed dependencies (**prominent at high and low wind speed, respectively**)."*

**Comment 2**

Section 5.1: This paragraph seems out of place. The wind direction analysis should be integrated with the mercury results rather than having its own section. The major result is the mercury analysis rather than the wind direction analysis.

**Response 2**

This section is meant to provide background and context for the Hg results. To improve the presentation we have moved this paragraph to section 2.2 and labeled it: "Meteorological Setting" as shown below (Please see the Line 180-184):

*2.2. Meteorological data*

*Hourly meteorological data (air temperature, relative humidity, and wind speed and direction)*

*were obtained from the Automatic Weather Station (AWS) operated by the Korea*
*Meteorological Administration (KMA) (http://www.kma.go.kr) (6 km from the site). Hourly*
*concentrations of $NO_2$, $O_3$, CO, $PM_{10}$ and $SO_2$ were obtained from the National Air Quality*
*Monitoring Network (NAQMN) (3 km from the site) (Fig. 1).*
*Meteorological Setting. Fig. S1 shows the frequency of counts of measured wind direction*
*occurrence by season during the sampling period. The predominant wind direction at the*
*sampling site was W (20.9%) and WS (19.2%), and calm conditions of wind speed less than 1*
*$m\ s^{-1}$ occurred 7.6% of the time. Compared to other seasons, however, the prevailing winds in*
*summer were N (17.0%), NE (16.4%), S (16.4%), and SW (15.8%).*

**Comment 3**
Line 293-294: "however considerably lower than those measured near large Hg sources in
Guangzhou, China (Table 1)." This is compared to a much older study. TGM in Guangzhou
from a more recent study in Table 1 was 4.6 ng/m3, which is similar to the average TGM in
this study.
**Response 3**
Thank you for this updated reference. We corrected **Table 1** on Line 561 and rephrased the
sentence as follows on Line 316-317.
*"…and those measured in China, in Japan and other locations in Korea, however lower than*
*those measured at Changchun, Gui Yang and Nanjing in China (Table 1)."*

**Comment 4**
Section 5.5 diurnal variations: The higher daytime than nighttime result is not quite correct. As
stated in lines 365-366, "TGM generally showed a consistent diurnal variation with an increase
in the early morning (06:00-09:00) and a decrease in the afternoon (14:00-17:00)." The daytime
period from 6:00-18:00 includes the morning increase and afternoon decrease; therefore it's
unclear if daytime TGM is really higher. I suggest explaining what caused the early morning
increase and afternoon decrease in TGM, instead of the cause for higher daytime TGM in
general. There were a few instances where it led to confusing results. E.g. line 384, "the higher concentrations in the daytime than those in nighttime were due to local emission sources
because the daytime temperature (14.7 ± 10.0 ℃) was statistically significantly higher than that
in the nighttime (13.0 ± 9.8 ℃) (t-test, $p < 0.05$) and there was a weak but statistically
significant negative correlation between TGM concentration and ambient air temperature ($r =$
$-0.08$) ($p < 0.05$)." The negative correlation between TGM and temperature is inconsistent with
the higher daytime TGM. Another example in line 406, "TGM concentration was negatively
correlated with ambient air temperature ($r = -0.08$) ($p < 0.05$) because high ambient air
temperature in the daytime will increase the height of the boundary layer and dilute the TGM,
and the relatively lower boundary layer at nighttime could concentrate the TGM in the
atmosphere (Li et al., 2011)." This explanation contradicts higher daytime TGM as well. It only
explains why TGM is lower in the afternoon, but not the early morning increase.
**Response 4**
To clarify this section the day-night concentration variation and morning-afternoon variation
during the day were discussed separately as shown below. Please see the Section 5.4 on Line
384-458.
*"Diurnal variations of TGM (Fig. 3), co-pollutants concentrations, and meteorological*
*data were observed (Fig. S4). TGM, $O_3$, CO, $SO_2$, and temperature in the daytime (06:00-*
*18:00) were higher than those in the nighttime (18:00-06:00) ($p < 0.05$) except $PM_{10}$ ($p =$*
*0.09) (Fig. S5). However, $NO_2$ during the nighttime because of relatively lower*
*photochemical reactivity with $O_3$ was higher than that in daytime ($p < 0.05$) (Adame et al.,*
*2012).*
*The daytime TGM concentration (5.3 ± 4.7 ng m$^{-3}$) was higher than that in the nighttime*
*(4.7 ± 4.7 ng m$^{-3}$) ($p < 0.01$), which was similar to several previous studies (Cheng et al.,*
*2014; Gabriel et al., 2005; Nakagawa, 1995; Stamenkovic et al., 2007) but different than*
*another studies (Lee et al., 1998). Previous studies reported that this different is due to local*
*sources close to the sampling site (Cheng et al., 2014; Gabriel et al., 2005), a positive*
*correlation between TGM concentration and ambient air temperature (Nakagawa, 1995) and*
*increased traffic (Stamenkovic et al., 2007). However, another study suggested that the*
*higher TGM concentration during the night was due to the shallowing of the boundary layer,*
*which concentrated the TGM near the surface (Lee et al., 1998).*

[revised manuscript text omitted]

CBPF for TGM concentrations since there were no correlations between TGM and other pollutants in spring, summer and fall except O3." I don't understand (and not mentioned in the paper) why the correlation results are needed to interpret the seasonal patterns of CPF and CBPF (Fig. S6). It wasn't needed to explain the overall CPF and CBPF results. Also, I'm still skeptical whether CBPF provides more information about sources than CPF. Wind speed dependency is included in CBPF to differentiate between ground level and stack emissions; however, it's not discussed in the results. There should be a more detailed discussion of the local ground level and stack emissions and uncertainties and disadvantages with the CBPF method.

**Response 5**

In order to clarify, we have deleted the following sentence.

*"It is difficult to discuss about the different seasonal patterns for CPF and CBPF for TGM concentrations since there were no correlations between TGM and other pollutants in spring, summer and fall except $O_3$."*

**Response to Anonymous Referee #2:**

**Comment 1**

Section 2 through 5 and conclusion are well improved. However, introduction section needs more work. Please re-organize and add recent literature reviews that are related with this research. Each paragraph contains one topic sentence. Please look at the introduction section. What is the topic sentence for each paragraph?

**Response 1**

Thank you for your comments. As suggested, we carefully revised and edited including adding recent literature reviews that are related with this research as follows on Line 71-80, Line 91-95, Line 109, Line 118-133 as shown below..

[revised manuscript text omitted]

**Comment 3**

The 3rd (line 85-92), the 4th (line 93-95), and 5th (line 96-103) paragraphs have same topics so they could be combined into one.

**Response 3**

As suggested, the sentences were combined into one paragraph on Line 91-112.

**Comment 4**

In the 6th paragraph (line 104-109), the author talked about the Hg emissions in South Korea.

Is this urban areas or both urban and rural areas? Which year? How about other counties near

South Korea? (e.g., China, Japan?)

**Response 4**

Kim et al. (2010) reported that the annual **average national** (including urban and rural areas)

anthropogenic Hg emissions from South Korea **in 2007** have been estimated to be 12.8

tons **ranged from 6.5 to 20.2 tons**.

In response to this comment, we have rephrased the sentence as follows on Line 113-114.

*"The annual average national anthropogenic Hg emissions from South Korea in 2007 have*

*been estimated to be 12.8 tons (range 6.5 to 20.2 tons);"*

We added the sentence about Hg emissions from China and Japan as follows on Line 118-122.

*"Global anthropogenic Hg emissions were estimated to be 1960 tons in 2010 with East and*

*Southeast Asia responsible for 777 tons (39.7%) (19.6 tons for Japan and 8.0 tons for South*

*Korea) (AMAP/UNEP, 2013). China is the largest Hg emitting country in the world,*

*contributing more than 800 tons (~ 40%) of the total anthropogenic Hg emissions (UNEP,*

*2008)."*

**Comment 5**

Line 117-121: should be one paragraph

**Response 5**

As suggested, we separated two paragraphs as follows on Line 142-146.

[revised manuscript text omitted]